# A self-supervised deep learning method for data-efficient training in genomics

Hüseyin Anil Gündüz[1,2], Martin Binder[1,2], Xiao-Yin To[1,2,3,4], René Mreches[3,4], Bernd Bischl[1,2], Alice C. McHardy [3,4], Philipp C. Münch [3,4,5,6,7✉] & Mina Rezaei [1,2,7✉]

Deep learning in bioinformatics is often limited to problems where extensive amounts of labeled data are available for supervised classification. By exploiting unlabeled data, self-supervised learning techniques can improve the performance of machine learning models in the presence of limited labeled data. Although many self-supervised learning methods have been suggested before, they have failed to exploit the unique characteristics of genomic data. Therefore, we introduce *Self-GenomeNet*, a self-supervised learning technique that is custom-tailored for genomic data. *Self-GenomeNet* leverages reverse-complement sequences and effectively learns short- and long-term dependencies by predicting targets of different lengths. *Self-GenomeNet* performs better than other self-supervised methods in data-scarce genomic tasks and outperforms standard supervised training with ~10 times fewer labeled training data. Furthermore, the learned representations generalize well to new datasets and tasks. These findings suggest that *Self-GenomeNet* is well suited for large-scale, unlabeled genomic datasets and could substantially improve the performance of genomic models.

[1] Department of Statistics, LMU Munich, Munich, Germany. [2] Munich Center for Machine Learning, Munich, Germany. [3] Department for Computational Biology of Infection Research, Helmholtz Center for Infection Research, 38124 Braunschweig, Germany. [4] Braunschweig Integrated Centre of Systems Biology (BRICS), Technische Universität Braunschweig, Braunschweig, Germany. [5] German Center for Infection Research (DZIF), partner site Hannover Braunschweig, Braunschweig, Germany. [6] Department of Biostatistics, Harvard School of Public Health, Boston, MA, USA. [7]These authors jointly supervised this work: Philipp C. Münch, Mina Rezaei. ✉email: philipp.muench@helmholtz-hzi.de; mina.rezaei@stat.uni-muenchen.de

In bioinformatics, using unlabeled data to augment supervised learning can reduce development costs for many machine learning (ML) applications that would otherwise require large amounts of annotation that are expensive to acquire, such as functional annotation of genes[1] or chromatin effects of single nucleotide polymorphisms. This is particularly the case in genomics due to the availability of large quantities of unlabeled sequence data from large databases and metagenomic studies.

In contrast to supervised methods, self-supervised learning (SSL) techniques learn representations that contain information about the properties of the data without relying on human annotation. The concept of SSL has been studied for several years in the field of ML. These SSL methods are unsupervised tasks, that are trained prior to the actual supervised training. Then, instead of training a supervised model from scratch, the representations learned from the SSL method can be used as a starting point for downstream supervised tasks such as taxonomic prediction or gene annotation. In this way, the pre-trained models can be used as a starting point that contains meaningful representations from the SSL tasks[2,3]. Several different methods for self-supervised representation learning have been proposed, e.g., in the fields of natural language processing (NLP)[4–8] and computer vision (CV)[3,9–11]. However, only a limited number of SSL methods have been developed for bioinformatics, and even fewer for omics data[12–15]. Thus, SSL has not yet seen such widespread adoption and remains an important and challenging endeavor in this field.

Existing methods for representation learning on omics-data have typically been adapted from other application fields of DL such as NLP or CV[16–18]. For example, DNA-Bert[19], which identifies conserved sequence motifs and candidate functional genetic variants, is an adaptation of BERT[2], which is a form of language model (LM)[20] that predicts masked tokens. These tokens are words in NLP, and nucleotides or k-mers in genome sequences. Contrastive-sc[18] is a method adapted from CV used for cell clustering based on single-cell RNA sequencing data. It creates two copies of each sequence with randomly masked nucleotides and then trains the network to maximize the agreement between the copies using a contrastive loss function, a method commonly used in CV[3]. CPCProt[17] is an adaptation of the contrastive predictive coding (CPC)[21] to protein data and is trained by predicting future amino acid sequence patches. However, there are specific properties of genome sequences that these methods do not take into account, resulting in non-optimal representations and limited use. Although used in several supervised methods, reverse-complement (RC) sequences have not been integrated into SSL methods. Additionally, nucleotides and k-mers contain low-content information compared to words in natural languages, and this is not taken into account when SSL methods developed for NLP are applied to genome data.

*Self-GenomeNet* overcomes these limitations. First, *Self-GenomeNet* uses RC sequences to create symmetry in the architecture. This increases predictive performance and reduces the number of model parameters. This may also have the desirable side effect of implicitly encoding RC-awareness in our architecture. Secondly, *Self-GenomeNet* predicts targets of different lengths as an SSL task. This way, a wider range of semantic relationships within the DNA data is learned. Finally, due to the way recurrent networks process their data, representations of many subsequences at different length scales are evaluated simultaneously within a single training step, leading to increased computational efficiency.

*Self-GenomeNet* makes more efficient use of unannotated genomic data to substantially improve various genomic tasks when limited labeled data is available, making it more suitable for genomic applications than existing SSL methods. In computational biology, such pre-training learning schemes could benefit a wide range of ML tools, where large amounts of unlabeled data such as metagenomic sequences are available to improve supervised models built on nucleotide-level training datasets.

## Results

***Self-GenomeNet* is an efficient self-supervised pre-training method, tailored for genomics.** *Self-GenomeNet* is a SSL method, where the network is trained without the need of labels on available sequential genome data. Then this network, particularly the trained weights of this network, can be used as the initial point of the model that will be trained for the supervised tasks, which are also called downstream tasks. We provide a model, which is trained on bacteria, virus, and human data without using labels. This model, named *generic Self-GenomeNet*, demonstrates robust performance across diversified tasks, providing researchers a readily accessible solution to leverage the power of our model in their own studies, particularly for their own supervised tasks. We have uploaded the trained generic *Self-GenomeNet* model to GitHub for easy access (*see self.genomenet.de*). Additionally, we have prepared interactive coding notebooks that provide detailed instructions on how to use this model to obtain embeddings of data and how to apply it to other datasets.

*Self-GenomeNet* learns representations of genome sequences through a defined pre-training task that does not require labels. This task is as follows: For a given input sequence of length $N$, $S_{1:N}$, an embedding of a subsequence $S_{1:t}$, predicts the embedding of the RC of the remaining subsequence $\bar{S}_{N:t+1}$. Thus, the model encodes in the learned representation of the given subsequence the essential information necessary to predict the RC of a neighboring subsequence. *Self-GenomeNet* encodes these two subsequences through a representation network consisting of a convolutional encoder network $f_\theta$, and a recurrent context network $C_\phi$ (Fig. 1a). The architecture of *Self-GenomeNet* is implemented to perform this prediction for multiple values of $t$ in one iteration, enabling a more efficient training procedure.

The network of *Self-GenomeNet* takes both $S_{1:N}$ and $\bar{S}_{N:1}$ as inputs. *Self-GenomeNet* encodes these two subsequences through a representation network consisting of a convolutional encoder network and a recurrent context network. As a result of the proposed architecture, the representations of subsequences $S_{1:t}$ and $\bar{S}_{N:t+1}$ are computed for multiple values of $t$ as intermediate outputs of the context network, while the whole sequences $S_{1:N}$ and $\bar{S}_{N:1}$ are encoded. Later, on top of the embedding representation, a linear prediction layer $q_\eta$ estimates the embedding of $\bar{S}_{N:t+1}$ from the embedding of $S_{1:t}$ using a contrastive loss against other random subsequences. Due to the symmetry of this design, $q_\eta$ is also used to predict the embedding of $\bar{S}_{N:t+1}$ from the embedding of $S_{1:t}$. Although only one prediction is shown in the figure for visual simplicity, the prediction is computed for multiple values of $t$. Contrastive loss is used for the optimization, meaning that the network is optimized so that the sequences (e.g., $S_{1:t}$ aims to predict the representation of the RC of its own neighbor (e.g., $\bar{S}_{N:t+1}$) among other representations in the training batch. The convolutional encoder network, the recurrent context network, and the linear prediction layer each consist of a single layer in our experiments to keep the architecture simple; however, more complex architectures are possible. The hyperparameters of the convolutional and recurrent networks are mentioned later in the paper, in the "Network Architecture Design" section.

After self-supervised training of the representations, these representations can be used for downstream supervised tasks by constructing a supervised deep learning model consisting of $f_\theta$ and $C_\phi$, followed by a fully connected output. The weights of $f_\theta$

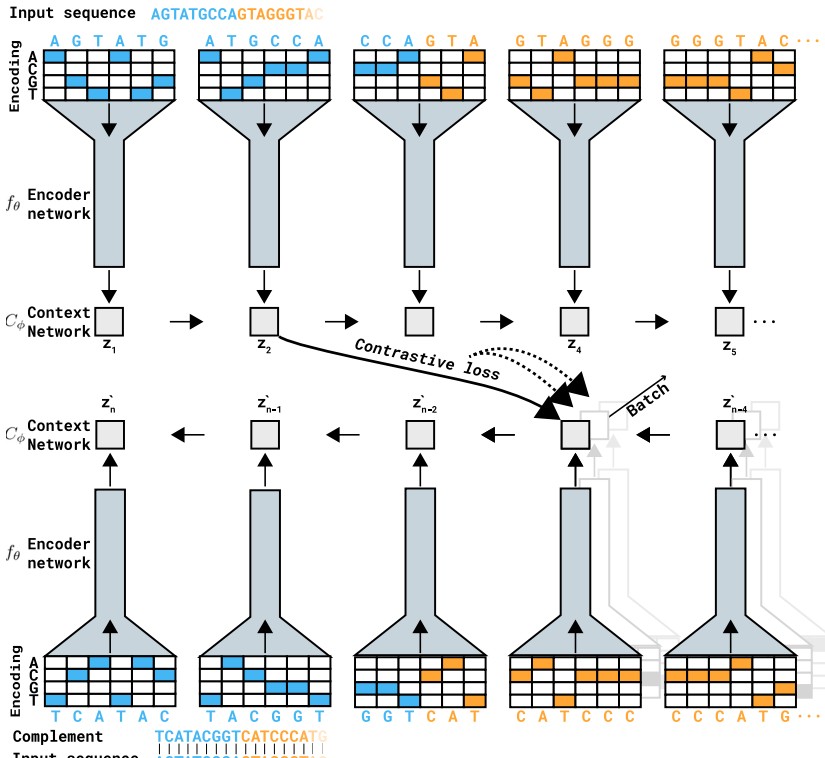

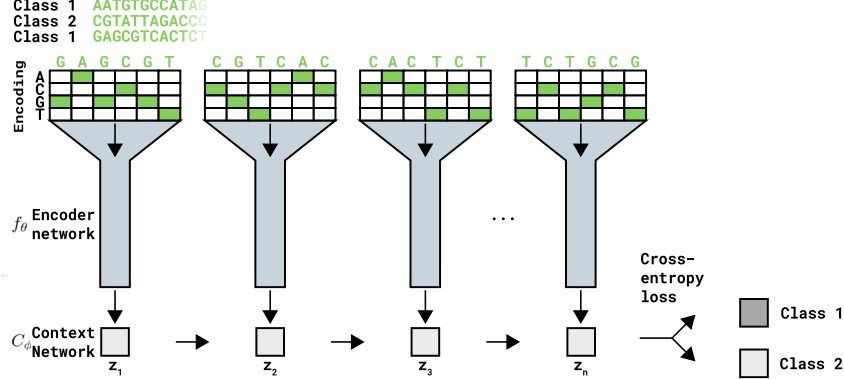

**Fig. 1 Pre-training of *Self-GenomeNet* and using the learned weights on a down-stream task. a** *Self-GenomeNet* takes part of a sequence as input and predicts the reverse-complement of the remaining sequence. The representations are learned by dividing unlabeled DNA sequences and their reverse-complements into patches, each of which is given as an input to an encoder network $f_\theta$. The outputs of $f_\theta$ are then fed sequentially to a recurrent context network $C_\phi$, resulting in representations of the input sequence up to a point $t$ ($S_{1:t}$) and representations of the reverse-complement of the input sequence going from ($t+1$) to the end (i.e., $\bar{S}_{N:t+1}$). The representations are computed for multiple values of $t$ simultaneously. Finally, the representations of $S_{1:t}$ ($z_i$) and $\bar{S}_{N:t+1}$ ($z'_{(n-1-i)}$) predict each other for multiple values of $t$ by using a contrastive loss, i.e, these sequences are matched among existing sequences in the training batch. Thus, in one iteration of the training of *Self-GenomeNet*, each of the computed representations $z_i$ and $z'_{(n-1-i)}$ are utilized efficiently since $z_i$ predicts $z'_{(n-1-i)}$ and $z'_{(n-1-i)}$ predicts $z_i$ for $i \in (1, 2, .., n-2)$ in one iteration of training. In the figure, we only show that $z_2$ predicts $z'_{(n-3)}$ for visual simplification. **b** The weights of $f_\theta$ and $C_\phi$ are initialized with the training results from the self-supervised learning task, but they are further trained (fine-tuned), along with the linear layer on the new supervised task.

and $C_\phi$ are initialized with the training results from the SSL task, but they are further trained (fine-tuned) together with the linear layer on the new supervised task (Fig. 1b).

We evaluated the performance of the representations obtained via *Self-GenomeNet* on different benchmarks (supervised tasks) using data of either viral, bacterial, or human origin: (i) The virus dataset contains viral genomes from GenBank[22] and RefSeq[23],

where the task is to classify prokaryotic viruses (bacteriophages) and eukaryotic viruses (termed "non-phages"). (ii) For bacterial data, we designed a supervised task on type VI secretion system identification (T6SS), where the task is to identify effector proteins among T6SS immunity proteins, T6SS regulators, and T6SS accessory proteins (SecReT6[24]). (iii) For the human dataset, we focus on the DeepSEA dataset[25]. The task is to classify 919

**Table 1 Experimental results in terms of class-balanced accuracy performance for design choices of Self-GenomeNet.**

| Target Sequences | | Pre-training Dataset / Dataset of Down-stream Task | | |
|---|---|---|---|---|
| Length | Targets | Virus / Virus (1000 nt.) | Bacteria / Virus | Bacteria / T6SS |
| Fixed Length | Reverse–Complement | 83.3 | - | - |
| Varying Length | Reverse | 87.8 | 76.9 | 70.2 |
| Varying Length | Forward | 85.8 | - | - |
| Varying Length | Reverse–Complement | 88.6 | 82.1 | 79.3 |

binary chromatin features such as transcription factor binding affinities, histone marks, and DNase I sensitivity.

We evaluated the performance of the representations obtained via *Self-GenomeNet* on different benchmarks (supervised tasks) using data of either viral, bacterial, or human origin: (i) The virus dataset contains viral genomes from GenBank[22] and RefSeq[23], where the task is to classify prokaryotic viruses (bacteriophages) and eukaryotic viruses (termed "non-phages"). The bacteriophage class contains approximately 1.0 billion nucleotides, the non-phage virus dataset ~0.5 billion nucleotides. (ii) For bacterial data, we designed a supervised task on type VI secretion system identification (T6SS), where the task is to identify effector proteins among T6SS immunity proteins, T6SS regulators, and T6SS accessory proteins (SecReT6[24]). This task is provided to demonstrate that our method works well on a dataset with real label scarcity, where the training set contains only 75 FASTA entries and ~0.3 million nucleotides. (iii) For the human dataset, we focus on the DeepSEA dataset[25]. It contains approximately 5 million subsequences of the human genome, with each sample containing 1000 nucleotides as input and a label vector for 919 binary chromatin features such as transcription factor binding affinities, histone marks and DNase I sensitivity. (iv) For the fungi-protozoa classification task, we downloaded DNAs of fungi and protozoa that may be pathogenic to humans from RefSeq[23]. Here, the training set contains approximately 2.7 billion nucleotides. (v) Finally, the bacteria data contains genomes from GenBank[22] and RefSeq[23], comprising ~83 billion nucleotides. It is used only for self-supervised pre-training.

In the results section, we will initially justify the choices we have made in our architectural design. First, we design an experiment to show the superiority of predicting targets of varying lengths over targets of fixed length. Then, we compare having the RC of neighboring subsequences as targets to be predicted with neighboring subsequences or their reverse. After justifying our design choices, we test *Self-GenomeNet* in data-scarce settings, where the labeled data is limited to a certain amount of the unlabeled data, and in transfer learning settings, where the pre-trained models are trained on different smaller datasets. Finally, we test *Self-GenomeNet* using the linear evaluation protocol[3,9,10,21,26–28], where the weights learned by self-supervision are frozen and thus not updated in the down-stream tasks. Here, only the fully connected layer on top of the frozen layers is trained. In these experiments, we compare *Self-GenomeNet* to four SSL baselines and the supervised baseline where the model is not pre-trained.

In all experiments except DeepSEA dataset, we report class-balanced accuracy and not precision/recall/F1 scores because these metrics put an emphasis on positive samples and also choosing a positive class. However, artificially choosing a positive class is harmful as detection of both classes holds equal importance for phage/non-phage classification and fungi/protozoa classification tasks. For the effector protein prediction task, assigning a positive class is also hard as the number of "effector protein" samples are more in both training, validation, and test set. For our experiments on the DeepSEA dataset, we

opted for average PR AUC as a metric, based on the findings of Quang and Xie[29], who demonstrated that the sparsity of positive binary targets in this dataset can artificially inflate the ROC AUC and thus PR AUC is a more suitable indicator of performance.

**Predicting the sequences of varying lengths improves the performance and has theoretical justifications.** *Self-GenomeNet* is a genome-tailored SSL method that aims to train meaningful representations for various genomic tasks by capturing the unique properties of genomic data. However, current models, such as LMs or CPC[21], use a fixed target sequence length, i.e., the part of the input sequence they are trying to predict has a constant size (up to 50 nucleotides). As supported by experiments comparing our method with these methods (Figs. 2, 3, and 4), training with such small subsequences does not yield optimal results, which may be because nucleotides and n-mers contain less information than words in natural languages. Optimal training for genomics also requires longer sequences with higher information content. Therefore, *Self-GenomeNet* predicts sequences of varying lengths, ranging from small to the maximum length of sequences that the model can capture.

*Self-GenomeNet* outperforms all other SSL methods (Figs. 2, 3, and 4) that predict sequences of fixed length, which may indicate the effectiveness of having targets of different lengths. However, we design an additional experiment that closely examines the effect of having target sequences of different lengths, which is unique to our model compared to the baselines. We show that having target sequences of different lengths helps the network to learn better representations. Thus, two self-supervised models are trained on the virus dataset with a sequence length of 1000 for the phage classification task. The first model, *Self-GenomeNet*, used varying-length targets as subsequences with a length range of $x \in \{40, 60, 80, ...960\}$, which predicted a subsequence with a length of $1000 - x$. The second model, also *Self-GenomeNet*, but with one modification, uses only two fixed-length sequences of 500 nt to predict each other, and all other settings are the same as the first model. Our evaluations show that predicting subsequences of varying lengths instead of fixed-length subsequences results in a considerable improvement in model accuracy. Specifically, the class-balanced accuracy on the test set increased from 83.3% to 88.6% when the weights learned through self-supervision (without using any labels) were frozen and only a fully connected layer was trained on top of these frozen layers using the same dataset as the downstream task (Table 1).

**Predicting the reverse-complement of the neighbor sequence improves the performance.** Most SSL methods were originally developed for NLP or CV tasks and did not consider the unique properties of genomic data. *Self-GenomeNet*, on the other hand, takes advantage of specific characteristics of genomic data by exploiting the fact that the reverse complement (RC) of a DNA sequence is also a valid DNA sequence. This allows for a symmetric construction of the SSL method, which reduces the

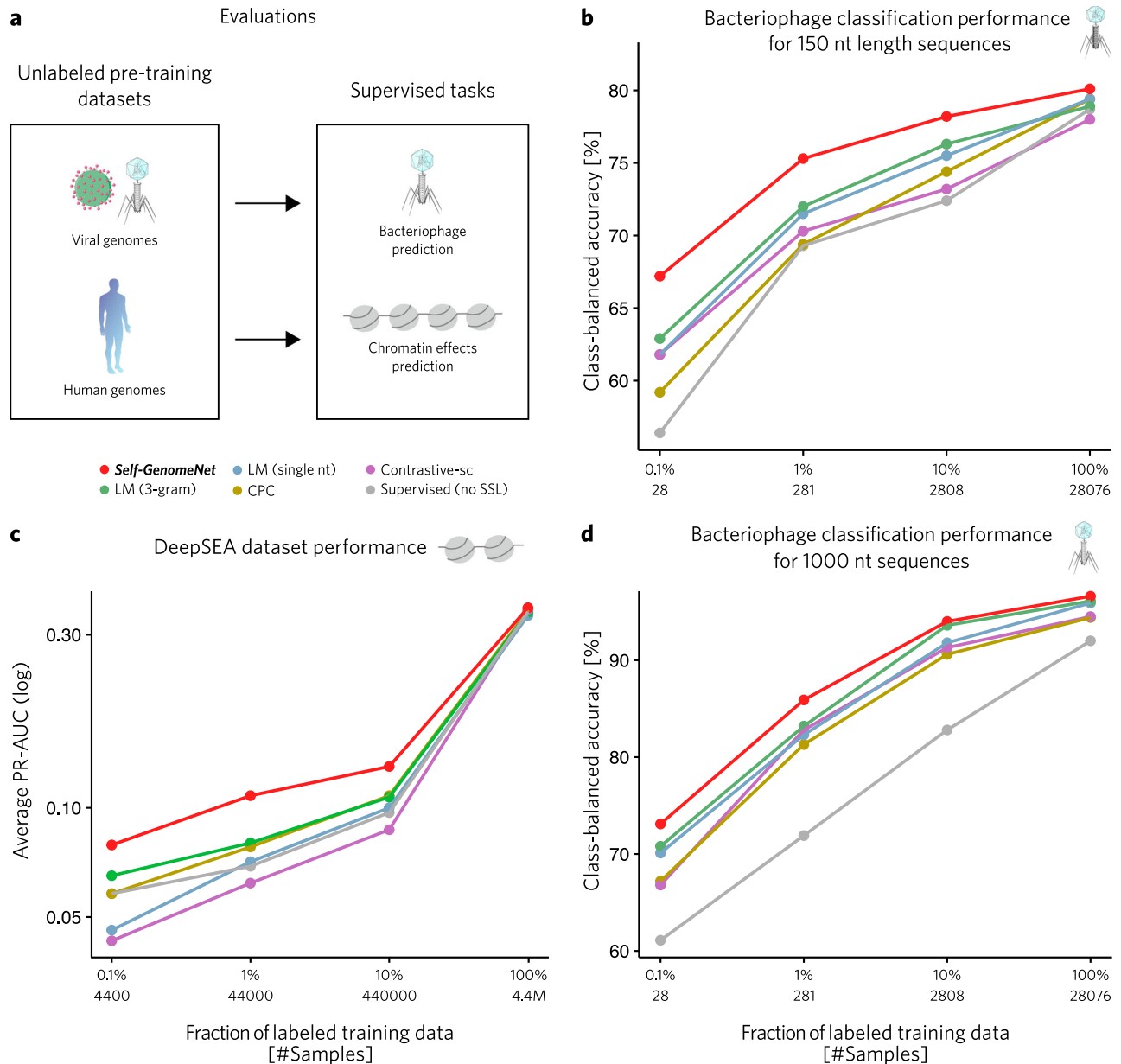

**Fig. 2 Comparison of self-supervised methods in data-scarce settings.** *Self-GenomeNet* representations outperform other baseline methods, such as language models[20] trained by predicting single nucleotides or 3-grams, Contrastive Predictive Coding[21], and Contrastive-sc[18], especially when a large fraction of available labels are omitted. We train the models in the datasets without using labels and then successively withhold labeled samples to mimic scenarios where labels are scarce (from 100% of available labeled samples to 0.1%). Each point in the plots is trained separately using the corresponding amount of labeled data. The weights of the context and encoder models are initialized with the training results from the SSL task, but they are trained further (fine-tuned), together with the linear layer, on the new supervised task. The label "Supervised" corresponds to the setting without any pre-training, where the weights are initialized randomly for the supervised task. **a** Overview of dataset and tasks used for evaluation. **b** The results of the viral dataset for 150 nt sequences, (**c**) the DeepSEA dataset, (**d**) and the viral dataset for 1000 nt sequences. The human icon representing the patient was created by Marcel Tisch and is available under a CC0 license. Original icon sourced from Bioicons. Twitter link for Marcel Tisch | CC0 License. The phage icon was created by DBCLS and is licensed under a CC-BY 4.0 Unported license. Modifications were made. Original icon sourced from Bioicons. DBCLS | CC-BY 4.0 License. The virus icon representing hepatitis was created by Servier and is licensed under a CC-BY 3.0 Unported license. Modifications were made. Original icon sourced from Bioicons. Servier | CC-BY 3.0 License. The chromatin structure icon was created by DBCLS and is licensed under a CC-BY 4.0 Unported license. Modifications were made. Original icon sourced from Bioicons. DBCLS | CC-BY 4.0 License.

number of model parameters and mitigates the risk of overfitting. Furthermore, we incorporate RC awareness into our learned representations by predicting RC targets.

We conducted an experiment to compare the effectiveness of different potential target sequences in self-supervised pre-training. Specifically, we compared the use of RC neighbor sequences $\bar{S}_{N:t+1}$ (which we refer to as "RC") with the use of

neighbor sequences $S_{t+1:N}$ (referred to as "Forward") and the reverse of neighbor sequences $S_{N:t+1}$ (referred to as "Reverse"). We examine both settings on a viral dataset (1000 nt) using the linear evaluation protocol[3,9,10,21,26–28], meaning that we freeze the weights trained on the viral dataset without using labels and then train a linear layer on top of these weights using the labels. We find that using RC targets results in a relative class-balanced

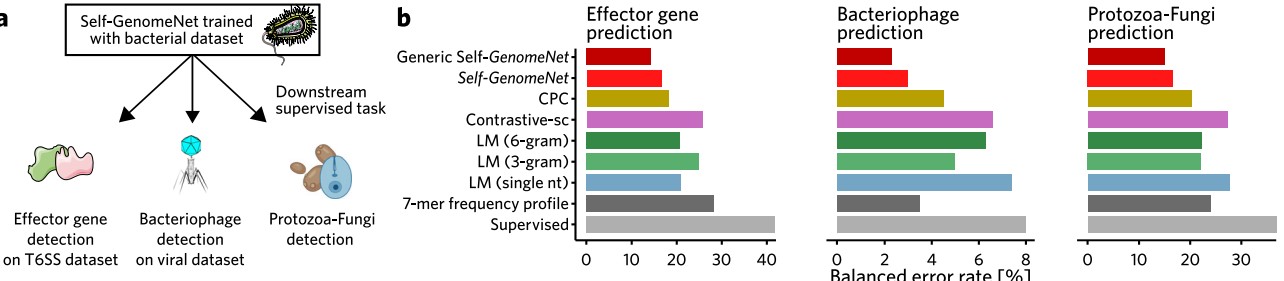

**Fig. 3 Comparison of self-supervised methods for transfer learning tasks.** *Self-GenomeNet* representations outperform other baseline methods, such as language models[20] trained by predicting single nucleotides, 3-grams or 6-grams, Contrastive Predictive Coding[21], and Contrastive-sc[18], when pre-trained with the bacteria dataset and then fine-tuned for effective gene detection and bacteriophage classification tasks. We also provide an additional evaluation, where we train *Self-GenomeNet* on a wider range of datasets, which includes bacteria, virus and human data (*generic Self-GenomeNet*). This model achieves even higher performance compared to *Self-GenomeNet*, showing that a wider range of data improves the performance of *Self-GenomeNet*. The context and encoder model weights are initialized with training results from the SSL task, but are further trained (fine-tuned) on the new supervised task along with an additional linear layer on top. The label "Supervised" and "7-mer frequency profile" corresponds to the setting without any pre-training, where the weights are randomly initialized for the supervised task. Here, the first model is the same architecture used in SSL settings, which similarly takes the one-hot encoded sequences. The second model is the CNN model developed by Fiannaca et al. [31], and it uses a 7-mer frequency profile as input. **a** Overview of the dataset and tasks used for evaluation. **b** The class-balanced accuracy performance for the effector gene detection task, the bacteriophage detection task, and for the protozoa-fungi prediction task. This figure was created in part with BioRender.com. The phage icon was created by DBCLS and is licensed under a CC-BY 4.0 Unported license. Modifications were made. Original icon sourced from Bioicons. DBCLS | CC-BY 4.0 License.

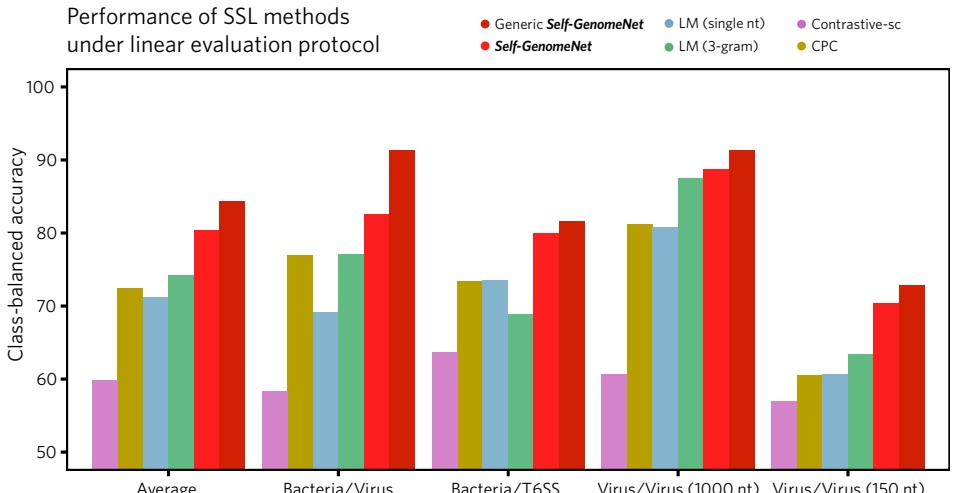

**Fig. 4 Comparison of self-supervised methods using the linear evaluation protocol.** Self-GenomeNet outperforms the baselines in all experiments using the linear evaluation protocol. First, the SSL methods are pre-trained on the bacteria and virus datasets. Then, the weights of the encoder and decoder networks learned in the pre-training are frozen and a linear layer on top of the model is trained on the T6SS and virus datasets. The relative increase in class-balanced accuracy over the second best-performing method is 9% on average, demonstrating the effectiveness of *Self-GenomeNet*. We also provide an additional evaluation, where we train *Self-GenomeNet* on a wider range of datasets, which includes bacteria, virus, and human data, which we call "*generic Self-GenomeNet*". This model achieves even higher performance compared to *Self-GenomeNet*, showing that a wider range of data improves the performance of *Self-GenomeNet*.

accuracy performance increase of 0.9% compared to using reverse targets and 3.3% compared to using forward targets (Table 1). We also discuss potential reasons for this performance change in the Discussion section.

Since the difference between RC and reverse targets was small in the experiment on the virus dataset, we investigate further. In the following experiments, we freeze the self-supervised weights trained on a bacteria dataset and train a linear layer on top of these weights for downstream tasks on the viral pre-training dataset, where the task is phage and non-phage classification and T6SS effector gene prediction. Our results show that using RC targets improved relative performance by 6.8% and 13.0% compared to using reverse sequences for these tasks, respectively. Therefore, our results suggest that using RC targets leads to better performance without increasing the number of parameters

(Table 1).

**Self-GenomeNet outperforms baselines in data-scarce settings, reducing the need for additional labeled data.** Generating large labeled datasets requires a substantial investment of resources that may not be feasible in computational biology. This limits the effectiveness of DL techniques. The use of unlabeled data is especially necessary when the labeled data is scarce since the accuracy of supervised DL models drops considerably in the low-data regime. We propose *Self-GenomeNet* as a data-efficient learning method to reduce the need for annotated data samples.

We mimic label scarcity scenarios by using the full datasets of virus and DeepSea datasets without labels and artificially reducing

the available labeled samples. Specifically, we test *Self-GenomeNet* by training on the full virus and DeepSEA datasets without using labels. We then quantify how model performance decreases as we successively withhold labeled samples to imitate scenarios where labels are scarce (from 100% of available labeled samples to 0.1%)[3,30]. Three different models are trained using *Self-GenomeNet* and baseline methods. Specifically, two models for 150 nt and 1000 nt long sequences are trained on virus data and one model is trained on the DeepSEA dataset. Then, each of these models is trained on data-scarce settings.

For the virus dataset in both cases of 150 nt (Fig. 2b) and 1000 nt (Fig. 2c) sequence length and the DeepSEA dataset (Fig. 2d), our method outperforms the four SSL baselines adapted from NLP or CV as well as the supervised baseline without any self-supervised training, at all evaluated fractions of available labels. We observe the most pronounced improvement in extreme data-scarce settings, such as 0.1% and 1%, with an average relative improvement of 11% and 14% over the second-best SSL method, respectively. *Self-GenomeNet* outperforms the supervised baseline that is trained with ten times more data (0.1% vs. 1%, 1% vs. 10%) on average across all experiments. This highlights that *Self-GenomeNet* representations are particularly effective in scenarios where labels are expensive to obtain—for example, in settings where genomic features must be manually validated in the lab.

**Learned representations of *Self-GenomeNet* can be transferred and adapted to new tasks and datasets**. We have quantified the transfer-learning capacity of representations trained on a large dataset of genome sequences to perform downstream supervised tasks on different and smaller genome datasets (Fig. 3)[3,30]. This evaluation particularly important because (i) for a given supervised downstream task, there may be little or no matching unlabeled training data (e.g., in the case of a newly discovered taxon, there may be no suitable training data available to train representations), and (ii) while performing the downstream task (training an arbitrary supervised model on top of the representations) is computationally fast, training the representation usually requires specialized hardware that is not available to many researchers.

For the transfer-learning tasks, we trained *Self-GenomeNet* and baseline models on a broad bacterial dataset. We then evaluated their predictive power on two tasks. First, we tested the transfer-learning ability of the pre-trained models on a very specific case: the effector gene prediction task, where we used the T6SS dataset[24]. The goal is to determine whether the pre-training regime works when the final supervised task is only a small subset of the self-supervised task. Next, we evaluated whether a biased training set for the pre-training task affects the prediction of the final supervised model. Here, the models (pre-)trained on the bacterial dataset (which might contain integrated prophages) are applied to the downstream tasks of separating bacteriophages from eukaryotic viruses.

*Self-GenomeNet* representations outperform those generated by the four competing baseline methods as well as the non-pretrained model. *Self-GenomeNet*, pre-trained on the bacteria dataset reduces the misclassification rate by 9% on the effector gene prediction task and by 33% on the phage identification task compared to the best performing SSL baseline. Compared to the non-pretrained baseline, the improvement on these tasks is as high as 60% and 63%, respectively. This shows that trained representations generalize well and are transferable to tasks with labeled data, even when this data differs from the self-supervised training data. This allows for applications where no suitable pre-training data is available.

**Self-GenomeNet outperforms all baselines in linear evaluation method**. We have shown that the representations learned by *Self-GenomeNet* exhibit transfer-learning capacity and that

*Self-GenomeNet* excels in data-scarce settings, consistently outperforming SSL baselines and supervised models over different fractions of available labels. However, all of the above evaluations include fine-tuning of the pre-trained weights (of the encoder and context networks). While the fine-tuning improves the performance in most of the cases, it makes it more difficult to evaluate the direct contribution of the SSL method due to the updated weights. To evaluate the quality of the embeddings learned by the SSL methods without making any modification on them, specifically by not fine-tuning them on downstream tasks, we use the linear evaluation protocol[3,9,10,21,26–28]. This method requires freezing the weights learned by self-supervision and thus not updating them in the downstream task, and training a fully connected layer is trained on top of the frozen layers on the downstream task. Therefore, the embeddings, which are the output of the pre-trained model, remain unchanged for a given input after the training on the downstream task. This simple and efficient method thus compares the effectiveness of SSL methods by directly comparing the embeddings themselves. In addition, training only the last linear layer is less computationally intensive than training the entire network, and achieving high performance without training the entire network may be useful for researchers with limited computational resources.

We performed this experiment and compared our method to the baselines on the virus dataset (phage classification task) for both 150 nt and 1000 nt, where the virus dataset is used for the pre-training and the downstream task. Additionally, we similarly froze the representations learned during the self-supervised pre-training on the bacteria dataset (for 1000 nt sequences) and evaluated the performance of these representations on the T6SS and virus datasets. We show that *Self-GenomeNet* outperforms the baselines in all experiments, and the relative increase in class-balanced accuracy over the second best method is 9% on average (Fig. 4).

## Discussion

We introduced *Self-GenomeNet*, a SSL technique designed specifically for genomic data. By leveraging RC sequences and predicting targets of different lengths, *Self-GenomeNet* overcomes the limitations of previous SSL methods and offers a more efficient use of unannotated genomic data. In our experiments, we compare *Self-GenomeNet* with several SSL baselines. We have shown that *Self-GenomeNet* outperforms CPC[21], which is potentially the most similar SSL method to ours, since both methods predict a target subsequence with contrastive loss. It also outperforms Contrastive-sc[18] as well as LMs[20] based on predicting single nucleotides and 3-grams. Both of these were originally proposed for CV and NLP, respectively, and have been applied in several cases in bioinformatics (Supplementary Methods, Supplementary Figs. 1, 2).

We have shown that *Self-GenomeNet* can also outperform supervised baselines that take normalized k-mer frequency as input. Specifically, we compare our model to the CNN model proposed by Fiannaca et al. [31]. The input of this model is a 7-mer frequency profile—the normalized frequency of 7-mers observed in the sequence. This input is fed into the model consisting of two convolutional layers with max-pooling layers, a flattened layer, and two fully connected layers. While this model requires an additional pre-processing step (in order to create the histogram based on 7-mers) and has approximately six times the number of parameters compared to our *Self-GenomeNet* model, our approach consistently outperforms this baseline in all experiments (Virus dataset (1000 nt) for phage/non-phage classification task, on T6SS dataset) (Fig. 3). Notably, *Self-Genomenet* archives substantially superior performance, particularly in data-scarce settings (Supplementary Fig. 4).

We tested the learned representations in the data-scarce regime, with only 0.1%, 1%, and 10% of the labeled data, and showed that *Self-GenomeNet* outperforms standard supervised training with only ~10 times less labeled training data. We tested the transfer learning capability and showed that our method achieves better performance compared to other SSL methods on our datasets. Finally, we show that the use of varying target lengths (based on a theoretical explanation) and the use of RC targets improve accuracy.

The effectiveness of the *generic Self-GenomeNet* model, which incorporates pre-training on virus, bacteria, and human data, consistently outperformed models pre-trained solely on virus data or bacteria data across all tested datasets on transfer learning tasks, under the linear evaluation protocol and across all proportions of labeled in data-scarce settings (Figs. 3, 4 and Supplementary Fig. 3). These results confirm the expected advantage of a network that is pre-trained on multiple sources of data compared to a network pre-trained on a single source. This aligns with the fundamental principles of ML, where leveraging diverse pre-training data often leads to improved performance.

Our study reveals an adaptation in SSL techniques that makes them well-suited for analysis of genomic datasets, allowing for more efficient use of genomic data. One adaptation is the use of the RC for data processing. Typically, DNA sequences are read from one end, but genes can be located on either strand of the DNA molecule. By accounting for RCs, *Self-GenomeNet* can efficiently learn powerful representations using the symmetry in the design. Processing both sequences with the same ML model and evaluating the average of the model's decisions in order to predict regulatory and taxonomic features is observed in several models in supervised training[32,33]. Therefore, feeding these two sequences to the same model such as a CNN and RNN model is a well-established practice for supervised learning tasks. However, the goal in these tasks is not to learn self-supervised representations, unlike our method, which to the best of our knowledge is the first method to use RC to make an SSL method more effective and efficient. On the other hand, these works justify the weight-sharing strategy in the encoder and context networks of our architecture.

Our study shows that using shared weights for the reciprocal prediction of two sequences, both of which are the RC of the upcoming subsequences for each other, improves the overall performance. While *Self-GenomeNet* predicts the representation of the RC of a neighboring subsequence $\overline{S}_{N:t+1}$ from $S_{1:t}$, the symmetry of the setup allows for also predicting $S_{1:t}$ from $\overline{S}_{N:t+1}$ and using shared weights for the whole model (prediction network, context network, and encoder network). Importantly, among the subsequence pairs that we can consider and study, only the subsequence pair $S_{1:t}$ and $\overline{S}_{N:t+1}$ can use the same encoder network, context network, and prediction network to predict each other, ensuring that the ML network performs the same task in both predictions. Specifically, we use these networks to predict the RC of the upcoming data for both predicting $S_{1:t}$ using $\overline{S}_{N:t+1}$ and predicting $\overline{S}_{N:t+1}$ using $S_{1:t}$. For other subsequence pairs we considered, which are $S_{1:t}$ and neighbor sequences $S_{t+1:N}$ (referred to as "Forward"), and $S_{1:t}$ and the reverse of neighbor sequences $S_{N:t+1}$ (referred to as "Reverse"), these networks do not have the same task, which results in a decrease in performance. Specifically, in the "Reverse" condition, context network with shared weights read these subsequences $S_{1:t}$ and $S_{N:t+1}$ in opposite directions and in the "Forward" condition the prediction networks with shared weights predict upcoming neighboring sequence when $S_{1:t}$ predicts $S_{t+1:N}$ and past neighboring subsequence when $S_{t+1:N}$ predicts $S_{1:t}$. The use of shared weights, which is possible by our proposed strategy of using the RC of the neighboring subsequences to predict each other,

reduces the number of learned parameters and the risk of overfitting and thus improves the performance.

*Self-GenomeNet*'s ability to reduce computation time by exploiting symmetry and RC is important for genomic research, where large datasets must be explored to gain insight into complex biological systems. *Self-GenomeNet* allows for efficient training by generating representations of multiple subsequence pairs simultaneously. Specifically, the representations of $S_{1:t}$ and $\overline{S}_{N:t+1}$, are computed for multiple values of $t$ in a single iteration. This is done by feeding both the input sequence $S_{1:N} = [s_1, s_2, \ldots, s_N]$ and the RC of that input $\overline{S}_{N:1} = [\overline{s}_N, \overline{s}_{N-1}, \ldots, \overline{s}_1]$ into the network, where $s_i \in \{A, C, G, T\}$ and $\overline{s}_i$ is the complementary nucleotide, e.g., $\overline{A} = T$. The network then evaluates representations of $S_{1:t}$ and $\overline{S}_{N:t+1}$ for multiple values of $t$ by design. We use all matching (neighbor) representations as pairs for self-supervised training. This leads to considerable efficiency in self-supervised pre-training. For example, in our experiments, there are 18 and 47 matching representations per each data sample in the batch respectively for 150 nt and 1000 nt sequences, respectively. Additionally, the number of predictions in an iteration is even double these values because the matching representations predict each other. All of these predictions are then used to optimize the model in one iteration (36 and 94 predictions, respectively) instead of having only one prediction per data sample in the batch, as is common with several other methods[3,18]. This makes *Self-GenomeNet* a computationally efficient SSL method.

Long short-term memory (LSTM) layers, which we use in our context network, are known to be less effective when they are fed inputs that contain much more time steps than 100[34]. Considering this, we designed our architectures to have 49 and 22 time steps fed into the context network, for our 1000 nt and 150 nt models, respectively. Specifically, we reduced the number of time steps by having a distance between the initial nucleotides of the created patches (Fig. 1) to be 20 and 6 respectively for these models. Having these values greater than 1 reduces the number of time steps considerably and using even greater values for this distance is recommended to be used for sequences that are much longer than 1000 nt. Therefore fairly limited short-term memory of LSTM can be managed. Additionally, it is also possible to change LSTM altogether with transformer-based models, which we will evaluate in the next version.

*Self-GenomeNet* has been shown to outperform other SSL methods in experiments with sequences of 150 and 1000 nt input lengths, demonstrating its effectiveness for sequences of varying lengths. Furthermore, *Self-GenomeNet* can be used to learn representations of sequences even longer than 1000 nt. However, pre-trained models that are trained on read-level sequences may not be effective for considerably longer sequences. Therefore, it may be necessary to pre-train a new model using *Self-GenomeNet* with longer sequences. It should be noted that training models on very long sequences can require a substantial amount of memory, making it difficult to fit many samples on a GPU. To ensure high batch size values, methods such as batch accumulation should be used instead of using a very small batch size (~10), which can negatively impact the effectiveness of self-supervised training. Therefore, practitioners should consider using batch accumulation when working with long sequences. Additionally, we suggest being cautious when interpreting results or masking low information sequences when the dataset contains a high repeat content such as transposable elements.

In our experiments, *Self-GenomeNet* showed resilience to changing architectural hyperparameters. Specifically, several architectural hyperparameters differ in the experiments with input length values of 150 nt and 1000 nt on the virus data, such as kernel and stride values of the convolutional layer. Despite the

different choices of hyperparameters, consistent improvements over other SSL methods are observed in both experiments, indicating the resilience of *Self-GenomeNet* to architectural changes. Therefore, we expect the performance of *Self-GenomeNet* to be superior to the other SSL methods for different model architectures. In addition, the performance of *Self-GenomeNet* can potentially be further enhanced by using different architectural modifications, such as deeper networks or alternative models for the encoder and context network, which we will evaluate in the next version.

In summary, our study shows that *Self-GenomeNet* can learn powerful representations from genomic datasets and can potentially be used to improve models trained on nucleotide-level data. Due to the improved performance in label-scarce settings, this method is of particular interest for the development of ML models, where the generation of labeled data is costly and only a limited number of labels are available. This could enable a new class of ML methods for domains such as functional prediction of genes, phenotypic or taxonomic prediction of genomes, or detection of loci of interest, such as pathogenicity islands.

## Methods

**Self-supervised training and contrastive loss in *Self-GenomeNet***. To create the representation, the sequence is first divided into $P$ overlapping patches $S_{p(j)}$ with the range $p(j) = (j \cdot a + 1) : (j \cdot a + l)$ indexing a subsequence, where $l$ is the patch length and $a$ the patch stride value, and $a < l$ for overlapping patches. The patches are first encoded using a convolutional neural network $f_\theta(\cdot)$. The resulting sequence of vectors $[f_\theta(S_{p(0)}), f_\theta(S_{p(1)}), \dots, f_\theta(S_{p(P-1)})]$ is fed into a recurrent context network $C_\phi(\cdot)$, yielding embeddings $z_i = C_\phi(\{f_\theta(S_{p(u)})\}_{u \le (i-(l/a))})$ for $(l/a) \le i < P + (l/a)$. The patches of the RC $\bar{S}_{\bar{p}(j)} = [\bar{s}_{j \cdot a + l}, \dots, \bar{s}_{j \cdot a + 1}]$ are also fed into $f_\theta$ first and then $C_\phi$, giving rise to $z_i$ is then a representation of $S_{1:i \cdot a}$ likewise, $\bar{z}_i$ represents $\bar{S}_{N:(i \cdot a + 1)}$.

Training the encoder and context networks consists of predicting $\bar{z}_i$ from $z_i$ contrastively against corresponding embeddings from other, negative example sequences $S^{(k)-}$, i.e., against $\bar{z}_i^{(k)-}$. This is done using a linear prediction layer $q_\eta$ and the Noise Contrastive Estimation or InfoNCE loss[35], which maximizes the mutual information shared between the forward sequence and its matching RC sequence, is used:

$$L_i = -\log \frac{\exp\left(\bar{z}_i^T q_\eta(z_i)\right)}{\exp\left(\bar{z}_i^T q_\eta(z_i)\right) + \sum_k \exp\left(\left(\bar{z}_i^{(k)-}\right)^T q_\eta(z_i)\right)} \quad (1)$$

Negative samples $\bar{z}_i^{(k)-}$ are efficiently generated by comparing against representations of other sequences loaded in the same mini-batch. Each sequence in the minibatch produces two negative samples for other sequences, the sequences themselves and their RC, resulting in $2(B-1)$ negative samples when using the minibatch size $B$.

Embeddings are always contrasted only against embeddings of sequences of the same length as the positive sample. More specifically, $z_i(S^+)$ predicts $\bar{z}_i(S^+)$ against $\bar{z}_i(S^{(k)-})$ where $S^{(k)-}$ contains $2(B-1)$ negative samples (other samples in the batch and their RCs) and not against $\bar{z}_{i+n}(S^-)$ where $n \ne 0$. This is done to prevent the network from learning to encode the represented length directly to gain an advantage, which would not be an intrinsically interesting feature for downstream tasks.

A loss term is introduced for each index $i$, denoting the number of patches represented by $z_i$. Due to the symmetry of the setup, the model both predicts $\bar{z}_i$ from $z_i$, as well as $z_i$ from $\bar{z}_i$. The

corresponding loss $\bar{L}_i$, induced by predicting $z_i$ from $\bar{z}_i$ then uses negative examples with the same length as $z_i$. The final loss for each individual sequence $S$ is thus defined by $L = \sum_i (L_i + \bar{L}_i)$.

**Self-GenomeNet maximizes the mutual information between varying-length targets and the representations, allowing the representations to effectively learn both short- and long-term information**. Our theoretical analysis illustrates the advantages of predicting sequences of different lengths. The mutual information between the predicted subsequences and the learned representations is maximized during self-supervised training[21]. However, optimizing the mutual information only for sequences that should contain limited long-range information may reduce the effectiveness of the learned representations because they may not capture important long-range information. Therefore, we propose a self-supervision method that maximizes the lower bound of the mutual information between the embeddings $z_i$ and varying-length RC targets $\bar{S}_{N:(i \cdot a + 1)}$ for $i \in \left\{ (l/a), ((l/a + 1)), \dots, ((N - l)/a) \right\}$, where $l$ is the length of each patch that is fed into the convolutional encoder network and $a$ the patch stride length, and $a < l$ for overlapping patches. Using the theoretical proof of CPC[21], we derive the theoretical derivation of the lower bound for maximizing the mutual information as follows: $I(\bar{S}_{N:(i \cdot a + 1)}, z_i) \ge \log(n) - L_i$, where $I(\bar{S}_{N:(i \cdot a + 1)}, z_i)$ is the mutual information between the learned representation $z_i$ and the RC of the consecutive subsequence $\bar{S}_{N:(i \cdot a + 1)}$. $n$ is the number of samples in the contrastive pre-training and is therefore $2(B-1) + 1 = 2B - 1$. The length of the predicted subsequence $\bar{S}_{N:(i \cdot a + 1)}$ changes as the value of $i$ changes, thereby allowing us to maximize the mutual information between targets of varying of length and the learned representations by optimizing the loss $L_i$ for different values of $i$ simultaneously. Moreover, this approach captures both short- and long-range semantics, and avoids the potential loss of long-range information that can occur when maximizing mutual information between learned representations and short patches that lack long-range information.

The theoretical derivation of the lower bound for maximizing the mutual information between targets of different lengths and the learned representations can be shown by adapting the equations in CPC[21] as follows:

Given that $\bar{S}_{N:(i \cdot a + 1)}$ is predicted from the representation $z_i$, our loss is given by Eq. (2).

$$L_i = -\mathbb{E}_S \left[ \log \frac{f\left(\bar{S}_{N:(i \cdot a + 1)}, z_i\right)}{\sum_{S_m \in S} f(S_m, z_i)} \right] \quad (2)$$

where $S$ includes $S^{(k)-}$ (the set containing all negative samples of the contrastive loss) and $\bar{S}_{N:(i \cdot a + 1)}$. $f$ is the modeled density ratio and is given by Eq. (3).

$$f\left(\bar{S}_{N:(i \cdot a + 1)}, z_i\right) = \exp\left(\bar{z}_i q_\eta(z_i)\right) \propto \frac{p\left(\bar{S}_{N:(i \cdot a + 1)}, | , z_i\right)}{p\left(\bar{S}_{N:(i \cdot a + 1)}\right)} \quad (3)$$

The optimal loss and a lower bound for this loss are then given by Eqs. (4) and (7) respectively.

$$L_i^{opt} = -\mathbb{E}_S \log \left[ \frac{\frac{p(\bar{S}_{N:(i \cdot a + 1)} | z_i)}{p(\bar{S}_{N:(i \cdot a + 1)})}}{\frac{p(\bar{S}_{N:(i \cdot a + 1)} | z_i)}{p(\bar{S}_{N:(i \cdot a + 1)})} + \sum_{S_m \in S^{(k)-}} \frac{p(S_m | z_i)}{p(S_m)}} \right] \quad (4)$$

$$= \mathbb{E}_S \log \left[ 1 + \frac{p\left(\bar{S}_{N:(i \cdot a + 1)}\right)}{p\left(\bar{S}_{N:(i \cdot a + 1)} | z_i\right)} \sum_{S_m \in S^{(k)-}} \frac{p(S_m | z_i)}{p(S_m)} \right] \quad (5)$$

$$\approx \mathbb{E}_S \log \left[ 1 + \frac{p\left(\bar{S}_{N:(i \cdot a+1)}\right)}{p\left(\bar{S}_{N:(i \cdot a+1)}|z_i\right)}(n-1)\mathbb{E}_{S_m}\frac{p(S_m|z_i)}{p(S_m)} \right] \quad (6)$$

$$= \mathbb{E}_S \log \left[ 1 + \frac{p\left(\bar{S}_{N:(i \cdot a+1)}\right)}{p\left(\bar{S}_{N:(i \cdot a+1)}|z_i\right)}(n-1) \right] \geq \mathbb{E}_S \log \left[ \frac{p\left(\bar{S}_{N:(i \cdot a+1)}\right)}{p\left(\bar{S}_{N:(i \cdot a+1)}|z_i\right)}n \right]$$
$$(7)$$

This is equal to $-I(\bar{S}_{N:(i \cdot a+1)}, z_i) + \log(n)$. Finally, we show that $I(\bar{S}_{N:(i \cdot a+1)}, z_i) \geq \log(n) - L_i$, where $I(\bar{S}_{N:(i \cdot a+1)}, z_i)$ is the mutual information between the learned representation $z_i$ and the RC of the consecutive subsequence $\bar{S}_{N:(i \cdot a+1)}$. $n$ is the number of samples in the contrastive pre-training and is therefore $2(B-1) + 1 = 2B - 1$. As the value of $i$ changes, the length of the predicted subsequence $\bar{S}_{N:(i \cdot a+1)}$ changes. We therefore maximize the mutual information between varying-length targets and the learned representations by optimizing the loss $L_i$ for different values of $i$ and simultaneously. Thus, the representations learn both short and long-range semantics. The higher performance of *Self-GenomeNet* compared to the baselines indicates that our strategy of maximizing the mutual information between varying-length subsequences and the learned representations is effective.

**Network architecture design**. The particular architectures of the encoder network $f_\Theta$ and the context network $C_\phi$ are hyperparameters of the method and can be chosen according to the task at hand. We choose $f_\Theta$ to be a convolutional layer with 1024 filters and $C_\phi$ to be an LSTM layer with 512 units. The kernel size of the convolutional layer is set equal to the patch size, which is the number of nucleotides given to the encoder network (Fig. 1a), and the stride value of the convolutional layer is equal to the distance between the starting points of the patches. For experiments trained on 150 nt sequences, the patch size is set to 24, and the stride is set to 6, resulting in 75% overlapping patches. For experiments trained on 1000 nt sequences, the patch size is set to 40 and the stride to 20, resulting in 50% overlapping patches.

**Model training process**. We use the Adam optimizer[36] with $\beta_1 = 0.9$, $\beta_2 = 0.999$ and a learning rate of 0.0001 for all experiments, except for the T6SS dataset fixed base network under transfer learning protocol, where the learning rate is set to 0.001 because we observe that 0.0001 is too low for this experiment. For weight initialization, Glorot uniform initialization[37] is chosen, which is the default Keras weight initialization. The size of the minibatch is chosen to be the largest possible for the used GPU, GeForce RTX 2080 Ti. Therefore, it is set to 128 for the self-supervised pre-training and 2048 for the supervised downstream tasks (only powers of 2 are considered). The hyperparameters, such as the hyperparameters of Adam[36] or learning rate, are set to the same values as our method for all baseline experiments. When some hyperparameters are unique to a baseline method, we follow the recommended values as in their papers.

In the transfer-learning experiments on the T6SS dataset and in the experiments with data-scarce settings, where 0.1% of the dataset is available, only the last linear layer of the model is trained with the labels as the first round of supervised training on downstream tasks. This means that the pre-trained layers are frozen at this stage, which is done to avoid rapid overfitting to the small labeled datasets, which results in low performance on the validation set. Then, in the second round of supervised training, the frozen layers are also fine-tuned, typically with considerably

fewer iterations than in the first round due to quick overfitting. In other transfer-learning and data-scarce experiments, the initial training with frozen layers is skipped for a faster evaluation process, as the preliminary experiments showed that it did not contribute to the final performance. Thus, the entire network is fine-tuned directly.

In the experiments in which the raw input data are fasta files (all datasets except DeepSEA), as long as the existing unlabeled data files are long enough, we generate sequences up to a certain number from the same fasta file for both supervised and self-supervised training. Thus, not only one data sample is generated when the fasta file is opened, as it would be if generated samples were completely random, in order to ensure much faster preprocessing. Specifically, up to 512 samples are created for fungi-protozoa dataset due to very long fasta files (and thus longer processing time) and 64 for other experiments. Additionally, this may help to create harder negative samples in the contrastive self-supervised training, which is shown to be helpful for learning better representations. However, hard-negative mining is not explicitly enforced in our experiments, such as by modifying the loss function[38]. While incorporating such measures can further improve the performance of *Self-GenomeNet*, the fact that we achieved robust results without relying on these underscores the robustness and success of our approach.

**Datasets**. The *DeepSEA* dataset[25] is an open benchmark dataset that has been evaluated by many other DL models[25,29,39]. It contains approximately 5 million subsequences of the human genome, with each sample containing 1000 nucleotides as input and a label vector for 919 binary chromatin features such as transcription factor binding affinities, histone marks, and DNase I sensitivity.

The *Virus* dataset is used as a representative of taxonomic classification tasks often encountered in metagenomics, where DNA found in environmental samples is analyzed by next generation sequencing[40]. We downloaded all publicly available viral genomes from GenBank[22] and RefSeq[23], and divided the dataset into two taxonomic classes of bacteriophages vs. viruses that are not bacteriophages, based on the annotations provided. Unlike DeepSEA, which identifies properties of genomic regions, this task tries to differentiate an aspect of an entire given genome sequence. We divided the downloaded FASTA files into training, validation, and test sets in approximate proportions of 70%, 20%, and 10%, respectively. The bacteriophage class contained approximately 1.0 billion nucleotides, the non-phage virus dataset ~0.5 billion nucleotides. Samples were created from FASTA files by partitioning them into equal-length non-intersecting sequences.

The *bacterial* dataset contains bacterial genomes from GenBank[22] and RefSeq[23], comprising approximately 83 billion nucleotides. It is used only for self-supervised pre-training. To create this dataset, we downloaded all publicly available bacteria genomes from GenBank, comprising approximately 83 billion nucleotides, and processed them similarly to how we processed the Virus dataset.

The *T6SS* effector protein dataset is provided to demonstrate that our method works well on a dataset with real label scarcity, where the training set contains only 75 FASTA entries. It is based on publicly available bacteria data (SecReT6[24]) where we defined the task as the identification of effector proteins. T6SS effector proteins serve as the positive samples to identify, whereas T6SS immunity proteins, T6SS regulators, and T6SS accessory proteins are negative samples. We divided the training, validation, and test sets into approximate proportions of 60%, 20%, and 20%, respectively.

For the fungi-protozoa classification task, we downloaded nucleotide data of fungi and protozoa that may be pathogenic to humans from RefSeq[23] using the genome_updater.sh script from https://github.com/pirovc/genome_updater with the parameters -g "fungi" -d "RefSeq" -c "representative genome" -A species:1 -a -p -T '4930,74721,4753,4827,5052,5475,5206,33183,5042,5151, 34487,4859' -k. For protozoa, we downloaded nucleotide information with the same script with the parameters -g "protozoa" -d "RefSeq" -c "representative genome" -m -A species:1 -a -p -T '554915,255975,5878,5794'. We divided the downloaded FASTA files into training, validation, and test sets in approximate proportions of 70%, 20%, and 10%, respectively.

**Reporting summary**. Further information on research design is available in the Nature Portfolio Reporting Summary linked to this article.

## Data availability
The data we used are publicly available and information about the data used is explained in the Datasets subsection. In addition, the training, validation and test sets can be found separately either as FASTA or RDS files, or as accession IDs on self.genomenet.de. Source data for figures can be found in Supplementary Data 1.

## Code availability
Code to reproduce and apply the models and pre-trained models are available via interactive notebooks under self.genomenet.de.

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

## Acknowledgements

We thank Curtis Huttenhower and Eric A. Franzosa for helpful discussions. This work was funded in part by the German Federal Ministry of Education and Research (BMBF) under Grant No. 01IS18036A and under GenomeNet Grant No. 031L0199A/031L0199B. P.C.M. received funding from the German Research Foundation (Grant number 405892038). X.-Y.T. received funding from the German Center for Infection Research (DZIF) TI BBD.

## Author contributions

The main idea of *Self-GenomeNet* is proposed by H.A.G. He developed and implemented the algorithm and he prepared the experimental results. X.-Y.T. has contributed to code development, mostly for the CPC baseline. R.M. also contributed to the code development, particularly regarding the processing of the data and language model baselines. M.B., B.B., and A.C.M. contributed to the writing of the manuscript. M.B. additionally contributed to the supervision of the project. The project is mainly supervised by M.R. and P.C.M. from a machine learning and bioinformatics perspective, respectively.

## Funding

## Competing interests

The authors declare no competing interests.
