## [Peer Review File · Communications Biology]

Reviewers' comments:

Reviewer #1 (Remarks to the Author):

Self-GenomeNet offers a self-supervised learning technique that can be applied to various deep learning tasks as a pre-trained base model. Unlike other techniques based on natural language processing, Self-GenomeNet takes into consideration the lack of context present within a DNA sequence. It can handle variable-length sequences through the application of a recurrent neural network. By its own nature, it does not require manual labeling of its training data and can train itself on a given genome. The authors compared the proposed network to four baselines and a supervised network on three datasets (viral, bacterial, and human).

Major:

- In the results, there was no evaluation of the embeddings themselves. What is the quality of these embeddings compared to ones produced by other DNA embedding techniques such as the one mentioned in kmer2vec: <https://doi.org/10.1089/cmb.2021.0536> or those generated by a variational auto encoder.
- From line 72 to line 75, these two sentences give the impression that the first part of the sequence is represented by one embedding and the reverse complement of the second half of the sequence is represented by one embedding. However, Figure 1 gives the impression that the two parts of the sequence are represented by a series of embeddings representing fix-sized short sub-sequences. The authors should clarify this point.
- In the paragraph starting a line 165 and Table 1, it is noted that predicting the reverse complement results in higher performance than in predicting the forward or the reverse strand. This finding is interesting, but the authors did not provide a reasonable explanation for this increase in performance.
- The architecture of the network, its input, its composing layers, its output, and an example of how it is applied should be mentioned in the very first paragraphs of the Result Section.
- The reviewer suggests additional experiments where this network is trained on DNA collected from several genomes across kingdoms, phylum, etc. We would like to see the results of a species-neutral network. Please evaluate this network on all of the mentioned datasets.
- The authors should compare to a method that is based on handling k-mer histograms; such methods can handle input sequences of variable lengths.
- The authors should also consider training this network to predict the embedding of a sequence given the sequence before and after, i.e., predicting the interleaving sub-sequence between two other sub-sequences, and compare to the proposed approach.
- The authors did not specifically discuss the cautiousness needed when training on genomes with high repeat content such as plant genomes and human genomes. Repeat sequences would dominate the training process.
- There was difficulty with following the paper in its current form.
- The authors note that Self-GenomeNet has difficulty with larger sequences. "LSTM and GRU... still have a fairly limited short-term memory, and they have a hard time learning long-term patterns in sequences of 100 time steps or more." (Hands-on Machine Learning with Scikit-learn, Keras & TensorFlow Page 520).

Minor:

- The details of the data sets should be expanded upon such as the size of the data set before discussing the results.
- The authors should explain what the linear evaluation protocol is, briefly, where it is mentioned.
- In Line 291–292, it is mentioned that one of the evaluation baselines uses 1 nucleotide and 3 grams. Do the 3 grams mean a trimer (3-mer)? If so, that is too simple of a baseline and is designed to perform poorly. It is better to evaluate against a pentamer or a hexamer.
- The authors should show the results of the network with fine-tuning and without fine-tuning in one figure for easier comparison.
- In Line 481, the subsection is not mentioned.

- In the discussion, the mathematical proof is mentioned very briefly at the beginning in the paragraph starting at Line 301. To improve the presentation of this paper, this paragraph should be moved to the Methods Section where they mentioned this proof in detail.
- In Line 141, accuracy is mentioned but not on which partition. Is it training, validation, or testing accuracy?

Reviewer #2 (Remarks to the Author):

Summary:

The paper introduces Self-GenomeNet, a self-supervised approach pre-trained on unlabeled data to learn robust representations using a contrastive learning framework. This network can then be used for downstream tasks using either linear probing, fine-tuning, or transfer learning based on the availability of training samples. The network is evaluated on different tasks in bacteria, virus, and human-centric datasets and performs well compared to the selected baselines. Code is provided, which will definitely help in the reproducibility of the work.

Overall impression:

I think the paper is well-written and makes a nice contribution to bringing advances in self-supervised learning from machine learning (ML), computer vision (CV), and natural language processing (NLP) communities to genomics. These approaches have been shown to have tremendously improved the capabilities of deep learning models in various applications. It is essential for the biology community to leverage these advances to accelerate the use of ML techniques in important problems. The extensive experiments have shown promise in the advances that can come from using these models. However, some details from the ML side (see next section for details) need to be critically considered and addressed to evaluate the contributions made in the proposed Self-GenomeNet framework adequately.

Specific comments for the authors to address:

1. How the data used for the pre-training stage differs from the target application is unclear. For example, I see a description of the data used for pre-training is primarily similar to the target task (Lines 221-237). While the two evaluation setups are constructed to assess the transferability of the tasks, I am not quite convinced whether the claim is sufficiently justified. For example, providing precision-recall curves for the predictions on tasks other than that found in the pretraining stage will provide a clearer picture of whether the representations are biased towards the distribution found in the pre-training data and whether it handles OOD data only if it is sufficiently different from pre-training. Similarly, I am not convinced about changing the data for pre-training the network for every experiment. The data is changed for experiments on bacteria (lines 229-237), viruses (lines 271-277), etc. In the ML/CV/NLP community, the standard practice is to fix the data for pre-training (it could come from different sources), pre-train the network and then evaluate its generalization (extrapolation if you want to call it that) capabilities on downstream tasks. The basic premise of self-supervised learning is not maintained by placing the constraint on what the data is pre-trained on. It would, by that definition, make this a semi-supervised approach where both labeled and unlabeled data are present, and the goal is to learn robust representation ****for that task****. This should be addressed to maintain the current protocols and terminologies used in the ML/CV/NLP community.
2. How is negative sampling done? Is it based on whether they are from a different read sequence? Or are some heuristics imposed on the data loading process to ensure hard negative mining, which is common in contrastive learning approaches? Several studies have shown that choosing positive and negative anchors is essential to the success of contrastive learning approaches. Some additional

details on what the data loading process will help with reproducibility.

3. Several approaches (a-d) have explored the use of self-supervised learning for genomics (albeit for sequence-level taxonomic classification from metagenomes) that, I feel, should be referenced to reflect the current state-of-the-art in ML/DL for genomics.

[a] Aakur, Sathyanarayanan N., et al. "Metagenome2Vec: Building Contextualized Representations for Scalable Metagenome Analysis." 2021 International Conference on Data Mining Workshops (ICDMW). IEEE, 2021.

[b] Indla, Vineela, et al. "Sim2real for metagenomes: Accelerating animal diagnostics with adversarial co-training." Advances in Knowledge Discovery and Data Mining: 25th Pacific-Asia Conference, PAKDD 2021, Virtual Event, May 11–14, 2021, Proceedings, Part I. Cham: Springer International Publishing, 2021.

[c] Aakur, Sathyanarayanan N., et al. "MG-NET: leveraging pseudo-imaging for multi-modal metagenome analysis." Medical Image Computing and Computer Assisted Intervention–MICCAI 2021: 24th International Conference, Strasbourg, France, September 27–October 1, 2021, Proceedings, Part V 24. Springer International Publishing, 2021.

[d] Queyrel, Maxence, et al. "Towards End-To-End Disease Prediction from Raw Metagenomic Data." International Journal of Biomedical and Biological Engineering 15.6 (2021): 234-246.

4. Ablation studies are not sufficiently backed up by quantitative evaluation. In lines 386-395, it was mentioned that changes in the hyperparameters of the network do not affect the performance of the approach. How much of the performance was affected? Is it an insignificant amount for ****all tasks****? If such a statement is made, more information on what was evaluated and its effect on the performance must be provided. I understand there is limited real estate in presenting the work, but this could be presented in the supplementary material. On that note, how were the hyperparameters selected? Was a grid search used? What was the range of values considered for the design?

5. There should be a discussion (maybe a short 1-2 lines) on the choice of evaluation metrics. Here, class-balanced accuracy is presented for all experiments. Why not precision/recall/F1 scores? In my opinion, when dealing with medical data in particular, minimizing false positives is more important than achieving greater (balanced) accuracy, which does not place more emphasis on false positives. I am sure that there is a reason for the choice of metric. Making it explicit will help understand the reasoning behind the approach and make it easier to evaluate the proposed model.

Responds to Reviewer 1

Q1: In the results, there was no evaluation of the embeddings themselves. What is the quality of these embeddings compared to ones produced by other DNA embedding techniques such as the one mentioned in kmer2vec: <https://doi.org/10.1089/cmb.2021.0536> or those generated by a variational autoencoder.

We appreciate the reviewer's comment and would like to clarify that the experiments conducted under the linear evaluation protocol in our study serve as a direct evaluation of the quality of the embeddings themselves. We understand that we may have not explicitly mentioned this in the results section and revised the manuscript to include a dedicated section discussing the evaluation of the embeddings.

This evaluation method we have used involves training a linear classifier on top of a frozen network that computes the embedding representations. The frozen network refers to the base model that is not fine-tuned in the downstream supervised task. Consequently, the embeddings generated by the pre-trained and not-fine-tuned (frozen) model are directly utilized for classification. We acknowledge that this is a common protocol for self-supervised learning evaluation to compare the quality of embeddings¹⁻⁷.

We now emphasized this more clearly by adding the following sentences in the Result Section:

“To evaluate the quality of the embeddings learned by the SSL methods without making any modification on them, specifically by not fine-tuning them on downstream tasks, we use the linear evaluation protocol¹⁻⁷. This method requires freezing the weights learned by self-supervision and thus not updating them in the downstream task, and training a fully connected layer is trained on top of the frozen layers on the downstream task. Therefore the embeddings, which are the output of the pre-trained model, remain unchanged for a given input after the training on the downstream task. This simple and efficient method thus compares the effectiveness of SSL methods by directly comparing the embeddings themselves.”

Furthermore, it is worth noting that the kmer2vec⁸ method mentioned by the reviewer is specifically designed to embed very short sequences such as 5-mers, 6-mers, and 7-mers, and the method is not designed to embed much longer sequences. In contrast, our self-supervised method aims to embed much longer sequences, such as sequences that are up to 1,000 nt long. As a result, a direct comparison between embeddings of a 1,000 nt long sequence and 7-mers would not be appropriate due to the significant difference in sequence length.

The variational autoencoders mentioned by the reviewer are typically employed for dimensionality reduction rather than for self-supervised representation learning, which focuses on achieving a higher accuracy on datasets with limited labeled data. Although there this an unpublished manuscript¹(link can be found as a footnote) that attempts to bridge the gap between self-supervised learning and variational autoencoders, it is important to note that the method has only been tested on computer vision tasks. Furthermore, the reported results in their manuscript indicate suboptimal performance, as mentioned by one of the reviewers.

Q2: From line 72 to line 75, these two sentences give the impression that the first part of the sequence is represented by one embedding and the reverse complement of the second half of the sequence is represented by one embedding. However, Figure 1 gives the impression that the two parts of the sequence are represented by a series of embeddings representing fix-sized short sub-sequences. The authors should clarify this point.

Thank you for bringing this to our attention. We apologize for any confusion caused by **Figure 1** and the corresponding sentence. We intentionally kept the figure simple in order to maintain clarity and avoid overwhelming visual complexity. We see that this may be misleading and we have revised the figure caption and provided a more detailed explanation as follows:

“Self-GenomeNet takes part of a sequence as input and predicts the reverse complement of the remaining sequence. The representations are learned by dividing unlabeled DNA sequences and their reverse complements into patches, each of which is given as an input to an encoder network f_θ . The outputs of f_θ are then fed sequentially to a recurrent context network C_ϕ , resulting in representations of the input sequence up to a point t ($S_{1:t}$) and representations of the reverse-complement of the input sequence going from $(t + 1)$ to the end (i.e., $\bar{S}_{N:t+1}$). The representations are computed for multiple values of t simultaneously. Finally, the representations of $S_{1:t}$ (z_i) and $\bar{S}_{N:t+1}$ ($z'_{(n-1-i)}$) predict each other for multiple values of t by using a contrastive loss, i.e, these sequences are matched among existing sequences in the training batch. Thus, in one iteration of the training of Self-GenomeNet, each of the computed representations z_i and $z'_{(n-1-i)}$ are utilized efficiently since z_i predicts $z'_{(n-1-i)}$ and $z'_{(n-1-i)}$ predicts z_i for $i \in (1, 2, \dots, n - 2)$ in one iteration of training. In the figure, we only show that z_2 predicts $z'_{(n-3)}$ for visual simplification.”

We have also made it clearer in the text to avoid any further confusion, as can be seen in the reply to the next point. For completeness, we also mention this text here:

¹ Self-Supervised Variational Auto-Encoders | OpenReview

“The network of Self-GenomeNet takes both $S_{1:N}$ and $S_{N:1}$ as inputs. Self-GenomeNet encodes these two subsequences through a representation network consisting of a convolutional encoder network and a recurrent context network. As a result of the proposed architecture, the representations of subsequences $S_{1:t}$ and $\bar{S}_{N:t+1}$ are computed for multiple values of t as intermediate outputs of the context network, while the whole sequences $S_{1:N}$ and $\bar{S}_{N:1}$ are encoded. Later, on top of the embedding representation, a linear prediction layer q_η estimates the embedding of $\bar{S}_{N:t+1}$ from the embedding of $S_{1:t}$ using a contrastive loss against other random subsequences. Due to the symmetry of this design, q_η is also used to predict the embedding of $\bar{S}_{N:t+1}$ from the embedding of $S_{1:t}$. Although only one prediction is shown in the figure for visual simplicity, the prediction is computed for multiple values of t . ”

Q3: The architecture of the network, its input, its composing layers, its output, and an example of how it is applied should be mentioned in the very first paragraphs of the Result Section.

Thank you for your feedback. We have made revisions to the Results section to address your concerns. In the very first paragraphs of the Result section, we now provide a more detailed description of the architecture, its input, composing layers, output, and an example of how it is applied, as follows:

“Self-GenomeNet is a self-supervised learning method, where the network is trained without the need of labels on available sequential genome data. Then this network, particularly the trained weights of this network, can be used as the initial point of the model that will be trained for the supervised tasks, which are also called downstream tasks. We provide a single model that demonstrates robust performance across diversified tasks, providing researchers a readily accessible solution to leverage the power of our model in their own studies, particularly for their own supervised tasks. We have uploaded the trained generic Self-GenomeNet model to GitHub for easy access (see <https://self.genomenet.de/>). Additionally, we have prepared interactive coding notebooks that provide detailed instructions on how to use this model to obtain embeddings of data and how to apply it to other datasets.

[...]

The network of Self-GenomeNet takes both $S_{1:N}$ and $\bar{S}_{N:1}$ as inputs. Self-GenomeNet encodes these two subsequences through a representation network consisting of a convolutional encoder network and a recurrent context network. As a result of the proposed architecture, the representations of subsequences $S_{1:t}$ and $\bar{S}_{N:t+1}$ are computed for multiple values of t as intermediate outputs of the context network, while the whole sequences $S_{1:N}$ and $\bar{S}_{N:1}$ are encoded. Later, on top of the embedding representation, a linear prediction layer q_η estimates the embedding of $\bar{S}_{N:t+1}$ from the

embedding of $S_{1:t}$ using a contrastive loss against other random subsequences. Due to the symmetry of this design, q_η is also used to predict the embedding of $\bar{S}_{N:t+1}$ from the embedding of $S_{1:t}$. Although only one prediction is shown in the figure for visual simplicity, the prediction is computed for multiple values of t . Contrastive loss is used for the optimization, meaning that the network is optimized so that the sequences (e.g. $S_{1:t}$) aims to predict the representation of the RC of its own neighbor (e.g. $\bar{S}_{N:t+1}$) among other representations in the training batch. The convolutional encoder network, the recurrent context network, and the linear prediction layer each consist of a single layer in our experiments to keep the architecture simple; however, more complex architectures are possible. The hyperparameters of the convolutional and recurrent networks are mentioned later in the paper, in the “Network Architecture Design” section [...]

Q4: In the paragraph starting a line 165 and Table 1, it is noted that predicting the reverse complement results in higher performance than in predicting the forward or the reverse strand. This finding is interesting, but the authors did not provide a reasonable explanation for this increase in performance.

We appreciate the reviewer's comment. In our revised manuscript, we have added a reference in the Results section to indicate that a more detailed explanation can be found in the Discussion section. The added reference reads as follows: “*We also discuss potential reasons for this performance change in the Discussion section.*”

Furthermore, in the Discussion section of the previously submitted version, we provide an explanation for the observed increase in performance. The related paragraph starts with: “*Our study shows that using shared weights for the reciprocal prediction of two sequences, both of which are the RC of the upcoming subsequences for each other, improves the overall performance...*”. We highlight the advantage of using shared weights for the reciprocal prediction of the reverse complement sequences, which allows the network to perform the same task in both predictions. This use of shared weights reduces the number of learned parameters and the risk of overfitting, thereby improving performance.

Additionally, we propose that the “Forward” condition may perform even poorer than the “Reverse” condition due to the inability to efficiently train the method using the representations of $S_{1:t}$ and $\bar{S}_{N:t+1}$ (Self-Genomenet - “RC”) or $S_{1:t}$ and $S_{N:t+1}$ (“Reverse”) for multiple values of t , as the representations of $S_{t+1:N}$ (“Forward”) cannot be computed for multiple values of t in a single iteration. This limitation affects the optimization process, as it is based on only one prediction per sequence for each iteration in the “Forward” condition. Although we believe this explanation is of minor importance compared to the performance difference between Self-GenomeNet and the other conditions, we can include it in the revised manuscript if requested.

Q5: The reviewer suggests additional experiments where this network is trained on DNA collected from several genomes across kingdoms, phylum, etc. We would like to see the results of a species-neutral network. Please evaluate this network on all of the mentioned datasets.

We appreciate the reviewer's suggestion to evaluate the network on DNA data collected from several genomes across kingdoms, phyla, etc., and to assess the performance of a "species-neutral" network. However, we would like to clarify our understanding of the reviewer's request to ensure we address it appropriately. Based on our understanding, we assume that the reviewer is asking how the inclusion of the data of diverse genomes (of more species) into the self-supervised training impacts the performance of our model. Previously, we had trained separate models on each of the three datasets (bacteria, virus, and human). We intentionally trained separate models on each of the three datasets (bacteria, virus, and human) to assess the generalizability of our approach using different pre-training datasets. This allowed us to evaluate the impact of the pre-training dataset's proximity to the downstream task and validate the effectiveness of our model. We believe that this approach provides valuable insights into the performance and robustness of our method in various scenarios.

However, in response to the reviewer's suggestion, we have now trained a model on the combined dataset, which includes DNA sequences from all these three classes: bacteria, virus, and human, labeled as generic *Self-GenomeNet*. We employed the Self-GenomeNet method to pre-train the model in a self-supervised manner, without relying on any labels. Subsequently, we fine-tuned this pre-trained model on the different tasks such as effector prediction, phage/non-phage detection task using all available labeled data for comparison with our baselines. Furthermore, based on a comment of **Reviewer 2 (Q1A)**, we have now added a new out-of-distribution benchmark dataset as a new transfer learning task for more thorough evaluation of the *generic Self-GenomeNet*. This new dataset comprises genomic sequences from fungi and protozoa, representing a distinct taxonomic group compared to our previous datasets. It focuses on the classification of fungi/protozoa that may be pathogenic to humans, based on DNA sequences downloaded from RefSeq⁹ (see **Methods**).

The results of our experiments demonstrate the effectiveness of the *generic Self-GenomeNet* model when compared to the bacteria-pretrained model. Specifically, the balanced-error-rate decreased from 16.6% to 15.0% in the fungi/protozoa classification task, from 16.6% to 14.3% in the effector protein detection task, and from 3.0% to 2.3% in the phage/non-phage task. These results indicate that the *generic Self-GenomeNet* model outperformed all other baselines and exhibited superior performance. We have updated **Figure 3** to include this generic *Self-GenomeNet* model as well as the new downstream supervised task.

This evaluation is now presented in the manuscript as follows:

“We provide a model, which is trained on bacteria, virus and human data without using labels. This model, named *generic Self-GenomeNet*, demonstrates robust performance across diversified tasks, providing researchers a readily accessible solution to leverage the power of our model in their own studies, particularly for their own supervised tasks. We have uploaded the trained *generic Self-GenomeNet* model to GitHub for easy access (see self.genomenet.de). Additionally, we have prepared interactive coding notebooks that provide detailed instructions on how to use this model to obtain embeddings of data and how to apply it to other datasets.”

Figure 3: Comparison of self-supervised methods for transfer learning tasks. *Self-GenomeNet* representations outperform other baseline methods, such as language models¹⁰ trained by predicting single nucleotides, 3-grams or 6-grams, Contrastive Predictive Coding³, and Contrastive-sc¹¹, when pre-trained with the bacteria dataset and then fine-tuned for effective gene detection and bacteriophage classification tasks. We also provide an additional evaluation, where we train *Self-GenomeNet* on a wider range of datasets, which includes bacteria, virus and human data (*generic Self-GenomeNet*). This model achieves even higher performance compared to *Self-GenomeNet*, showing that a wider range of data improves the performance of *Self-GenomeNet*. The context and encoder model weights are initialized with training results from the SSL task, but are further trained (fine-tuned) on the new supervised task along with an additional linear layer on top. The label “Supervised” and “7-mer frequency profile” corresponds to the setting without any pre-training, where the weights are randomly initialized for the supervised task. Here, the first model is the same architecture used in SSL settings, which similarly takes the one-hot encoded sequences. The second model is the CNN model developed by Fiannaca et al., and it uses a 7-mer frequency profile as input. This model is also 6 times bigger in terms of number of parameters. a) Overview of the dataset and tasks used for evaluation. b) The class-balanced accuracy performance for the effector gene detection task, the bacteriophage detection task, and for the protozoa-fungi prediction task.

Further, we have conducted experiments to compare the *generic Self-GenomeNet* model to other SSL models under linear evaluation protocol, where the weights learned by self-supervision are frozen and thus not updated in the downstream tasks. Here, only the fully connected layer on top of the frozen layers are trained. In comparison to *Self-GenomeNet*, which is pre-trained only on one dataset (either virus or bacteria, as indicated on the plot for different cases), the *generic Self-GenomeNet* model, pretrained on the combined data, demonstrates improvements in class-balanced accuracy in both downstream tasks of phage/non-phage detection (virus) and effector gene prediction (T6SS) and for both sequence length of 1000 nt and 150 nt on the virus dataset. Specifically, we observed the following performance improvements:

Figure 4: Comparison of self-supervised methods using the linear evaluation protocol. *Self-GenomeNet* outperforms the baselines in all experiments using the linear evaluation protocol. First, the SSL methods are pre-trained on the bacteria and virus datasets. Then, the weights of the encoder and decoder networks learned in the pre-training are frozen and a linear layer on top of the model is trained on the T6SS and virus datasets. The relative increase in class-balanced accuracy over the second best-performing method is 9% on average, demonstrating the effectiveness of *Self-GenomeNet*. We also provide an additional evaluation, where we train *Self-GenomeNet* on a wider range of datasets, which includes bacteria, virus and human data, which we call “*generic Self-GenomeNet*”. This model achieves even higher performance compared to *Self-GenomeNet*, showing that a wider range of data improves the performance of *Self-GenomeNet*.

Further, we have conducted experiments to compare the *generic Self-GenomeNet* model to the model pre-trained solely on virus data. We tested these models with varying proportions of labeled data available for training, including 0.1%, 1%, 10%, and 100% of the labeled data (**Supplementary Figure 3**). In comparison to *Self-GenomeNet*, which is pre-trained only on virus data, the generic *Self-GenomeNet* model, pretrained on the combined data, demonstrated improvements in class-balanced accuracy across different proportions of labeled data. Specifically, we observed the following performance improvements:

The introduction of labeled data had a significant impact on the class-balanced accuracy in our experiments. With 0.1% labeled data, we observed an increase from 73.1% to 74.7%. When the labeled data increased to 1%, the class-balanced accuracy improved even further, from 85.9% to 89.0%. The largest improvement was seen with 10% labeled data, where the accuracy rose from 94.0% to 96.3%. Finally, with 100% labeled data, the class-balanced accuracy increased from 96.6% to 97.7%. These results underline the importance and effectiveness of labeled data in improving classification accuracy.

Supplementary Figure 3: Data-scarce settings performance comparison on the virus dataset. Generic Self-GenomeNet and Self-GenomeNet trained only on this dataset are evaluated. The generic *Self-GenomeNet*, which incorporates pre-training on virus, bacteria, and human data consistently outperformed the model pre-trained solely on virus data (the whole dataset) across all proportions of labeled data-scarce settings. We train the models without using labels and then successively withhold labeled samples of the viral dataset for 1,000 nt sequences to mimic scenarios where labels are scarce (from 100% of available labeled samples to 0.1%). Each point in the plots is trained separately using the corresponding amount of labeled data. The weights of the context and encoder models are initialized with the training results from the SSL task, but they are trained further (fine-tuned), together with the linear layer, on the new supervised task.

The effectiveness of the generic *Self-GenomeNet* model, which incorporates pre-training on virus, bacteria, and human data, is highlighted by these findings. It consistently outperformed models pre-trained solely on virus data across all tested proportions of labeled data.

We also added this finding to the Discussion section of the manuscript:

“The effectiveness of the generic Self-GenomeNet model, which incorporates pre-training on virus, bacteria, and human data consistently outperformed models pre-trained solely on virus data or bacteria data across all tested datasets on transfer learning tasks, under the linear evaluation protocol and across all proportions of labeled in data-scarce settings (Fig. 3, Fig. 4, Supplementary Fig. 3). These results confirm the expected advantage of a network that is pre-trained on multiple sources of data compared to a network pretrained on a single source. This aligns with the fundamental

principles of machine learning, where leveraging diverse pre-training data often leads to improved performance.”

If our understanding of the reviewer's request is not accurate, we kindly request clarification so that we can address it properly in the next round of revisions.

Q6: (This was indicated as a minor issue, but we reply here in connection to the previous question) In Line 291–292, it is mentioned that one of the evaluation baselines uses 1 nucleotide and 3 grams. Do the 3 grams mean a trimer (3-mer)? If so, that is too simple of a baseline and is designed to perform poorly. It is better to evaluate against a pentamer or a hexamer.

We appreciate your thoughtful comment and understand your concern about the potentially oversimplified 3-gram baseline. It is important to note that the baseline does not predict only one 3-mer but rather predicts multiple 3-mers throughout the sequence. For example, in the case of a 1,000 nt long sequence, a total of 66 3-mers are predicted, resulting in the prediction of 198 nts in total. You can also find the schematic of these methods in the supplementary material (see **Supplementary Fig. 2**). While we acknowledge that using higher-order k-mers such as pentamers or hexamers could provide additional information, it is important to consider the trade-off between complexity and computational feasibility.

In response to your recommendation, we have incorporated a 6-gram model into our evaluations, and found that the 3-gram baseline still serves as a reasonable comparison point for assessing the performance of our proposed method.. Across the reported experiments, on average over three different tasks indicated in this figure, we observe that the 6-gram model performed slightly better than or comparably to 3-gram model (**Fig. 3**). For instance in the effector gene prediction task, it outperformed the 3-gram model and achieved similar results to the single nucleotide model. In the phage classification task, the 6-gram model performed worse than the 3-gram model but better than the single nucleotide model. Similarly, in the new protozoa-fungi classification task, the 6-gram model showed comparable performance to the 3-gram model and outperformed the single nucleotide model (**Fig. 3**). This suggests that the additional complexity introduced by the 6-gram model did not lead to substantial improvements in performance compared to the 3-gram model. Considering this finding, it can be argued that the 3-gram model is not too simple for the task at hand. While higher-order k-mer models might capture more intricate patterns, the marginal performance gains observed with the 6-gram model suggest that the additional complexity may not be necessary for the specific tasks evaluated. The 3-gram model, which captures local patterns in DNA sequences, still provides a meaningful baseline for comparison and demonstrates competitive performance. Therefore, based on the comparison with 6-gram we can conclude that the 3-gram model is a reasonable choice and not considered too simple, given that the more complex 6-gram model did not significantly outperform it.

However, it is important to emphasize that our proposed method, *Self-GenomeNet*, consistently outperformed all language models across all tasks (**Fig. 2, 3, 4**). While the different k-mer models exhibited varying degrees of performance, our self-supervised approach demonstrated superior results.

Q7: The authors should compare to a method that is based on handling k-mer histograms; such methods can handle input sequences of variable lengths.

Thank you for your suggestion. We have added a further comparison to our method as a supervised baseline, specifically the CNN-model of “Deep learning models for bacteria taxonomic classification of metagenomic data”¹². This model takes normalized number of k-mer occurrences as input, or equivalently normalized frequency of k-mers observed in the sequence.

We added this text to the Discussion section of the manuscript:

“We have shown that *Self-GenomeNet* can also outperform supervised baselines that take normalized k-mer frequency as input. Specifically, we compare our model to the CNN model proposed by Fiannaca et al¹². The input of this model is a 7-mer frequency profile - the normalized frequency of 7-mers observed in the sequence. This input is fed into the model consisting of two convolutional layers with max pooling layers, a flatten layer, and two fully connected layers. While this model requires an additional pre-processing step (in order to create the histogram based on 7-mers) and has approximately six times the number of parameters compared to our *Self-GenomeNet* model, our approach consistently outperforms this baseline in all experiments (Virus dataset (1,000 nt) for phage/non-phage classification task, on T6SS dataset) (**Fig. 3**). Notably, *Self-Genomenet* archives significantly superior performance, particularly in data-scarce settings (**Supplementary Fig. 4**).”

Figure 3: Comparison of self-supervised methods for transfer learning tasks. *Self-GenomeNet* representations outperform other baseline methods, such as language models¹⁰ trained by predicting single nucleotides, 3-grams or 6-grams, Contrastive Predictive Coding³, and Contrastive-sc¹¹, when pre-trained with the bacteria dataset and then fine-tuned for effective gene detection and bacteriophage classification tasks. We also provide an additional evaluation, where we train *Self-GenomeNet* on a wider range of datasets, which includes bacteria, virus and human data (*generic Self-GenomeNet*). This model achieves even higher performance compared to *Self-GenomeNet*, showing that a wider range of data improves the performance of *Self-GenomeNet*. The context and encoder model weights are initialized with training results from the SSL task, but are further trained (fine-tuned) on the new supervised task along with an additional

linear layer on top. The label “Supervised” and “7-mer frequency profile” corresponds to the setting without any pre-training, where the weights are randomly initialized for the supervised task. Here, the first model is the same architecture used in SSL settings, which similarly takes the one-hot encoded sequences. The second model is the CNN model developed by Fiannaca et al., and it uses a 7-mer frequency profile as input. This model is also 6 times bigger in terms of number of parameters. a) Overview of the dataset and tasks used for evaluation. b) The class-balanced accuracy performance for the effector gene detection task, the bacteriophage detection task, and for the protozoa-fungi prediction task.

Supplementary Figure 4: Data-scarce Settings Performances of generic Self-GenomeNet and a supervised model that takes 7-mer profile as input. The generic *Self-GenomeNet* outperforms the CNN model proposed by Fiannaca et al¹². *The input of this model is a 7-mer frequency - the normalized frequency of 7-mers observed in the sequence. While this model requires an additional pre-processing step (in order to create the histogram based on 7-mers) and has approximately six times the number of parameters compared to our Self-GenomeNet model, our approach consistently outperforms this baseline, particularly in data-scarce settings. We train the generic Self-GenomeNet using virus, bacteria, and human data without using labels and then successively withhold labeled samples of the viral dataset for 1,000 nt sequences to mimic scenarios where labels are scarce (from 100% of available labeled samples to 0.1%). Each point of the generic Self-GenomeNet in the plots is trained separately using the corresponding amount of labeled data. The weights of the context and encoder models are initialized with the training results from the SSL task, but they are trained further (fine-tuned), together with the linear layer, on the new supervised task. The label “7-mer frequency profile” corresponds to the setting where we used the CNN model proposed by Fiannaca et al, where the weights are initialized randomly for the supervised task.*

Q8: The authors should also consider training this network to predict the embedding of a sequence given the sequence before and after, i.e., predicting the interleaving sub-sequence between two other sub-sequences, and compare to the proposed approach.

Thank you for your valuable suggestion. We have carefully considered training the network to predict the embedding of a sequence given the sequence before and after, as you recommended. However, integrating this self-supervised task into our current framework poses several challenges that may compromise these advantages provided by our approach

Firstly, our method is specifically designed to compute tens or hundreds of representations of varying-length sequences in a single iteration, enhancing optimization efficiency and effectiveness (paragraph starting with “*Self-GenomeNet’s* ability to reduce computation time ..” in Discussion Section). Implementing the suggested method would likely require multiple iterations or additional computational steps, which could adversely impact the computational advantages offered by our approach. More precisely, it is not possible to create embeddings of sequences with varying length for both two predicting and one predicted sequence in the suggested method at one iteration, which is a key advantage our proposed Self-GenomeNet offers, as mentioned in, for example, the following text among some other texts in the manuscript: “*As a result of the proposed architecture, the representations of subsequences $S_{1:t}$ and $\bar{S}_{N:t+1}$ are computed for multiple values of t as intermediate outputs of the context network, while the whole sequences $S_{1:N}$ and $\bar{S}_{N:1}$ are encoded.*”.

Secondly, our architecture utilizes a recurrent network, which inherently captures directional information. To incorporate the proposed method, we would need to employ two separate recurrent networks to handle forward and reverse sequences. This would increase the number of parameters and introduce complexity to the network, deviating from the simplicity and parameter sharing. that our architecture aims to maintain. Our experiments comparing Self-GenomeNet (also same as “Reverse-Complement” condition) against the “Fixed Length,” “Reverse,” and “Forward” conditions demonstrate the importance of parameter sharing (discussed in detail in paragraph starting with “Our study shows that using shared weights for ...” in Discussion Section) induced by our method (**see Table 1**).

Thirdly, implementing the proposed method would likely require modifications to the loss function as two embeddings would be used to predict one embedding, which would also introduce additional complexity to the network architecture.

Lastly, it is important to consider that the suggested method may not provide additional insights into why our method works, such as showing the significance of having varying-length targets and parameter sharing. In contrast, comparing Self-GenomeNet

against "Fixed Length," "Reverse," and "Forward" conditions allows us to gain insights into the effectiveness of our approach.

We sincerely appreciate the thoughtful suggestion from the reviewer, and we hope that our explanation clarifies why integrating the proposed method into our framework will not be advantageous.

Q9: The authors did not specifically discuss the cautiousness needed when training on genomes with high repeat content such as plant genomes and human genomes. Repeat sequences would dominate the training process.

Thanks for pointing this out, one issue when training models on genomic data is that there are repeat contents such as tandem repeats or interspersed repeats such as Alu elements. In the human genome, up to 50% of the genome has unknown function or, sometimes referred to as the "dark matter". It was found that half of this consists of repetitive sequences¹³. These repeats are often assumed to provide no information for the particular tasks and can be considered as noise. Since deep learning methods are generally known to be stable to such noise or low quality datasets as discussed in Rolnick et al.¹⁴, there is no extensive work that checks this influence for genomic datasets, however, based on our experiments in other studies, deep learning methods seem powerful to not be biased for these repeats.

We acknowledge the potential issue raised regarding the influence of noise during pre-training, specifically when genomes with varying repeat levels are used for downstream tasks. It is indeed conceivable that the classifier may leverage the degree of noise for prediction. However, we have not observed this in our experiments.

Furthermore, it's important to note that even sequences considered as "noise" or "junk" DNA may contain valuable information. It is often challenging to distinguish between noise and meaningful information in the context of genomic sequences. Recent studies suggest that these repetitive elements can influence the evolution of genomes, particularly by modulating gene activity. For example, it seems that these elements have an influence on the evolution of genomes, particularly by controlling gene activity¹⁵. Therefore, even if the classifier were leveraging these elements, it might be learning meaningful biological signals rather than merely capitalizing on noise.

One solution would be to mask or remove regions in the training dataset that is low complex or similar to repeats. However, our benchmarking shows that even when not doing masking, the Self-GenomeNet method learns representations that are helpful for predicting the downstream task. Further, masking could introduce systematic bias that might mislead the neural network by picking up the signal.

We have added this to the discussion section which reads:

“We suggest being cautious when interpreting results or masking low information sequences, when the dataset contains a high repeat content such as transposable elements.”

Q10: There was difficulty with following the paper in its current form.

Thanks for pointing this out. In the rebuttal process, we have further clarified several paragraphs, figure titles and technical terms based on comments of both reviewers. We kindly ask you to go through our modifications in the manuscript, in which we indicated the modified text with blue color. If you have further recommendations, we are happy to be in discussion regarding how we can improve the manuscript and incorporate the changes if necessary.

Q11: The authors note that Self-GenomeNet has difficulty with larger sequences. “LSTM and GRU... still have a fairly limited short-term memory, and they have a hard time learning long-term patterns in sequences of 100 time steps or more.” (Hands-on Machine Learning with Scikit-learn, Keras & TensorFlow Page 520).

Thank you for bringing up this point. For such sequences some architectural changes could be made to make Self-GenomeNet more efficient, such as reducing the number of time steps per nt by increasing the distance between the initial nucleotides of the patches (see **Fig. 1**). Therefore fairly limited short-term memory of LSTM could be managed. We added the text below to the Discussion Section:

*“LSTMs, which we use in our context network, are known to be less effective when they are fed inputs that contain much more time steps than 100¹⁶. Considering this, we designed our architectures to have 49 and 22 time steps fed into the context network, for our 1000 nt and 150 nt models, respectively. Specifically, we reduced the number of time steps by having a distance between the initial nucleotides of the created patches (**Fig. 1**) to be 20 and 6 respectively for these models. Having these values greater than 1 reduces the number of time steps substantially and using even greater values for this distance is recommended to be used for sequences that are much longer than 1000 nt. Therefore fairly limited short-term memory of LSTM can be managed. Additionally, it is also possible to change LSTM altogether with transformer-based models, which we will evaluate in the next version. “*

Q12: The details of the data sets should be expanded upon such as the size of the data set before discussing the results.

We now modified the following text in the Results section by carrying information from the Dataset section, which the size of the data is mentioned:

“We evaluated the performance of the representations obtained via Self-GenomeNet on different benchmarks (supervised tasks) using data of either viral, bacterial, or human

origin: (i) The virus dataset contains viral genomes from GenBank¹⁷ and RefSeq⁹, where the task is to classify prokaryotic viruses (bacteriophages) and eukaryotic viruses (termed “non-phages”). The bacteriophage class contains approximately 1.0 billion nucleotides, the non-phage virus dataset approximately 0.5 billion nucleotides. (ii) For bacterial data, we designed a supervised task on type VI secretion system identification (T6SS), where the task is to identify effector proteins among T6SS immunity proteins, T6SS regulators, and T6SS accessory proteins (SecReT6¹⁸). This task is provided to demonstrate that our method works well on a dataset with real label scarcity, where the training set contains only 75 FASTA entries and approximately 0.3 million nucleotides. (iii) For the human dataset, we focus on the DeepSEA dataset¹⁹. It contains approximately 5 million subsequences of the human genome, with each sample containing 1000 nucleotides as input and a label vector for 919 binary chromatin features such as transcription factor binding affinities, histone marks and DNase I sensitivity. (iv) For the fungi-protoczoa classification task, we downloaded fungi and protozoa data that may be pathogenic to humans from RefSeq⁹. Here, the training set contains approximately 2.7 billion nucleotides. (v) Finally, the bacteria data contains genomes from GenBank¹⁷ and RefSeq⁹, comprising approximately 83 billion nucleotides. It is used only for self-supervised pre-training.”

Q13: The authors should explain what the linear evaluation protocol is, briefly, where it is mentioned.

We added more detail about linear evaluation protocol in Results Section to make it clearer for the reader:

“We test Self-GenomeNet using the linear evaluation protocol, where the weights learned by self-supervision are frozen and thus not updated in the downstream tasks. Here, only the fully connected layer on top of the frozen layers are trained.”

Similarly, we provide more explanations on the Results Section, “Predicting the reverse-complement of the neighbor sequence improves the performance” subsection:

“We examine both settings on a viral dataset (1.000 nt) using the linear evaluation protocol, meaning that we freeze the weights trained on the viral dataset without using labels, and then train a linear layer on top of these weights using the labels.”

Q14: The authors should show the results of the network with fine-tuning and without fine-tuning in one figure for easier comparison.

Thanks for the suggestion. The reason for having them on different plots is that they are different evaluation protocols and the motivation behind these protocols are also different. For example, one motivation of the linear evaluation protocol, which we now highlighted based on your recommendation, is that this protocol provides a direct comparison of the learned representations (embeddings), which is not the case for the

fine-tuned models. Additionally, as we have run experiments on multiple supervised downstream tasks, having them on the same plot would likely make the plots visually overwhelming and it would also be harder to convey the motivation of the figure and the implications of the results.

Q15: In Line 481, the subsection is not mentioned.

Sorry, it was a typo, which we fixed in the revision.

Q16: In the discussion, the mathematical proof is mentioned very briefly at the beginning in the paragraph starting at Line 301. To improve the presentation of this paper, this paragraph should be moved to the Methods Section where they mentioned this proof in detail.

Thanks for the suggestion. We have moved this paragraph to the Methods Section where we mentioned this proof in detail.

Q17: In Line 141, accuracy is mentioned but not on which partition. Is it training, validation, or testing accuracy?

It is the testing accuracy. Now we emphasized this as follows: “*Specifically, the class-balanced accuracy on the test set increased from 83.3% to 88.6% [...]*”.

Reviewer 2

Q1A: How the data used for the pre-training stage differs from the target application is unclear. For example, I see a description of the data used for pre-training is primarily similar to the target task (Lines 221-237). While the two evaluation setups are constructed to assess the transferability of the tasks, I am not quite convinced whether the claim is sufficiently justified. For example, providing precision-recall curves for the predictions on tasks other than that found in the pretraining stage will provide a clearer picture of whether the representations are biased towards the distribution found in the pre-training data and whether it handles OOD data only if it is sufficiently different from pre-training.

We believe that the T6SS dataset¹⁸ and the phage/non-phage classification task^{9,17} differ sufficiently from the pre-training dataset, which is bacteria data^{9,17}.

While there may be some genomes that exist in both the pre-training dataset, which contains bacteria data^{9,17}, and the supervised training datasets, which are the T6SS dataset¹⁸ and the phage/non-phage classification data^{9,17} these are not deliberately included in both. More importantly, the datasets used for supervised training are in essence significantly different from self-supervised dataset, such as in terms of data distribution. As a result, although there is likely an overlap between pre-training and the supervised downstream training datasets, it is expected to be limited.

Having a similarity at this level between a self-supervised pre-training dataset and supervised downstream tasks is in fact not unusual for the machine-learning community. For example, well-known papers in self-supervised learning like SimCLR⁵ and BYOL²⁰, among others²¹ pre-train their self-supervised learning models on the ImageNet dataset, which includes, for example, images of pets, cars and flowers. These self-supervised models pre-trained on ImageNet²² are commonly evaluated on datasets like pets²³, flowers²⁴, or cars²⁵ as transfer learning tasks. Thus, we argue that it is a common practise for the ML community to evaluate the pre-trained models on transfer learning tasks and datasets that have this level of similarity to the pre-training datasets.

However, to further address your concern, we have followed your recommendation and included a new dataset and task in our study. This dataset consists of samples containing DNAs of fungi and protozoa⁹ that can be pathogenic to humans and the task involves classifying them based on DNA sequences, specifically focusing on the fungi/protozoa classification. We have added these information to the manuscript, specifically to the Results Section and to the Methods (Datasets) Section respectively:

“For the fungi-protozoa classification task, we downloaded DNAs of fungi and protozoa that may be pathogenic to humans from RefSeq⁹. Here, the training set contains approximately 2.7 billion nucleotides.”

“For the fungi-protozoa classification task, we downloaded nucleotide data of fungi and protozoa that may be pathogenic to humans from RefSeq²⁴ using the genome_updater.sh script from https://github.com/pirovc/genome_updater with the parameters -g "fungi" -d "refseq" -c "representative genome" -A species:1 -a -p -T '4930,74721,4753,4827,5052,5475,5206,33183,5042,5151,34487,4859' -k. For protozoa, we downloaded nucleotide information with the same script with the parameters -g "protozoa" -d "refseq" -c "representative genome" -m -A species:1 -a -p -T '554915,255975,5878,5794'. We divided the downloaded FASTA files into training, validation, and test sets in approximate proportions of 70%, 20%, and 10%, respectively.”

The results can be found in **Fig. 3.**, which you may also find in our reply to **Q5** of the **Reviewer 1**. The results demonstrate that *Self-GenomeNet* also outperforms the baselines in this benchmark. More information on *generic Self-GenomeNet* is provided in our reply to your concern in **Q1B**.

Q1B: Similarly, I am not convinced about changing the data for pre-training the network for every experiment. The data is changed for experiments on bacteria (lines 229-237), viruses (lines 271-277), etc. In the ML/CV/NLP community, the standard practice is to fix the data for pre-training (it could come from different sources), pre-train the network and then evaluate its generalization (extrapolation if you want to call it that) capabilities on

downstream tasks. The basic premise of self-supervised learning is not maintained by placing the constraint on what the data is pre-trained on. It would, by that definition, make this a semi-supervised approach where both labeled and unlabeled data are present, and the goal is to learn robust representation **for that task****. This should be addressed to maintain the current protocols and terminologies used in the ML/CV/NLP community.**

We appreciate your valuable feedback and the opportunity to clarify the motivation behind changing the pre-training dataset. One of our key motivations was to assess the generalizability of our approach by using different data sources for pre-training and different evaluation tasks and benchmark how the similarity of the pre-training dataset to the downstream task influence the accuracy.

In response to this suggestion and to the suggestion from Reviewer 1, we have added a *generic Self-GenomeNet* model that incorporates data from multiple sources, including human, bacteria, and virus. Regarding the modifications we have made regarding your request, we kindly ask you to read our rebuttal to the **Q5** of the **Reviewer 1**. This allowed us to further investigate the performance and effectiveness of our approach when trained on a more diverse set of genomes.

Q2. How is negative sampling done? Is it based on whether they are from a different read sequence? Or are some heuristics imposed on the data loading process to ensure hard negative mining, which is common in contrastive learning approaches? Several studies have shown that choosing positive and negative anchors is essential to the success of contrastive learning approaches. Some additional details on what the data loading process will help with reproducibility.

In the experiments, in which the raw input data are fasta files, as long as the existing unlabeled data files are long enough, we generate sequences up to a certain number from the same fasta file, for both supervised and self-supervised training. The exception to this is the DeepSEA dataset, where the raw input data do not consist of fasta files and the predetermined size of the full DNA sequences is exactly 1,000 nt, unlike fasta files, where the typical length of the sequences are much longer.

We generate multiple files from the same fasta file mainly because the processing of the data samples are much faster this way, i.e. processing a fasta file (for example opening it) takes long time, and using only one sequence from a fasta file (to make it completely random) makes training much slower. However, several selected negative samples may effectively be much harder than randomly selected negative samples in the self-supervised training, since the negative samples generated from the same fasta file, in which the positive sample is also generated, are likely to be more similar to the positive sample. Thus it may be harder for the model to choose the positive sample in the batch during the contrastive learning, which may mean that we utilise implicit hard negative mining.

Although we may have utilized hard negative mining implicitly, we have not explicitly enforced the hard-negative mining, such as by modifying the loss function. However, our method already demonstrated significant effectiveness even without incorporating such a measure. The fact that we achieved robust results without relying on these additional techniques further underscores the robustness and success of our approach. Nevertheless, we believe the performance can be further enhanced by using explicit hard-negative mining, and thus can be considered as a future work.

We also added the following text to the Methods Section, "Model Training Process" Subsection:

"In the experiments, in which the raw input data are fasta files (all datasets except DeepSEA), as long as the existing unlabeled data files are long enough, we generate sequences up to a certain number from the same fasta file, for both supervised and self-supervised training. Thus, not only one data sample is generated when the fasta file is opened, as it would be if generated samples were completely random, in order to ensure much faster preprocessing. Specifically, up to 512 samples are created for fungi-protocista dataset due to very long fasta files (and thus longer processing time) and 64 for other experiments. Additionally, this may help to create harder negative samples in the contrastive self-supervised training, which is shown to be helpful for learning better representations. However, hard-negative mining is not explicitly enforced in our experiments, such as by modifying the loss function²⁶. While incorporating such measures can further improve the performance of Self-GenomeNet, the fact that we achieved robust results without relying on these underscores the robustness and success of our approach."

Q3: Several approaches (a-d) have explored the use of self-supervised learning for genomics (albeit for sequence-level taxonomic classification from metagenomes) that, I feel, should be referenced to reflect the current state-of-the-art in ML/DL for genomics.

[a] Aakur, Sathyanarayanan N., et al. "Metagenome2Vec: Building Contextualized Representations for Scalable Metagenome Analysis." 2021 International Conference on Data Mining Workshops (ICDMW). IEEE, 2021.

[b] Indla, Vineela, et al. "Sim2real for metagenomes: Accelerating animal diagnostics with adversarial co-training." Advances in Knowledge Discovery and Data Mining: 25th Pacific-Asia Conference, PAKDD 2021, Virtual Event, May 11–14, 2021, Proceedings, Part I. Cham: Springer International Publishing, 2021.

[c] Aakur, Sathyanarayanan N., et al. "MG-NET: leveraging pseudo-imaging for multi-modal metagenome analysis." Medical Image Computing and Computer Assisted Intervention–MICCAI 2021: 24th International Conference, Strasbourg, France, September 27–October 1, 2021, Proceedings, Part V 24. Springer International Publishing, 2021.

[d] Queyrel, Maxence, et al. "Towards End-To-End Disease Prediction from Raw Metagenomic Data." International Journal of Biomedical and Biological Engineering 15.6 (2021): 234-246.

We have now added the mentioned papers to the second paragraph of the Introduction Section.

Q4: Ablation studies are not sufficiently backed up by quantitative evaluation. In lines 386-395, it was mentioned that changes in the hyperparameters of the network do not affect the performance of the approach. How much of the performance was affected? Is it an insignificant amount for **all tasks****? If such a statement is made, more information on what was evaluated and its effect on the performance must be provided. I understand there is limited real estate in presenting the work, but this could be presented in the supplementary material. On that note, how were the hyperparameters selected? Was a grid search used? What was the range of values considered for the design?**

In the lines you mentioned, we mean that we conducted experiments on the viral dataset for data-scarce settings and linear evaluation for two different values of sequence length: sequences of both 150 nt and 1,000 nt lengths. We used different sets of hyperparameters for each case in order to design a more optimal architecture based on expert knowledge - as the main reason is indicated in the added text to the manuscript (below in quotation marks). Despite these variations, Self-GenomeNet consistently outperformed the baseline methods, indicating its potential to achieve superior performance across different hyperparameter settings.

We have now added the following paragraph to the manuscript:

*“LSTMs, which we use in our context network, are known to be less effective when they are fed inputs that contain much more time steps than 100¹⁶. Considering this, we designed our architectures to have 49 and 22 time steps fed into the context network, for our 1000 nt and 150 nt models, respectively. Specifically, we reduced the number of time steps by having a distance between the initial nucleotides of the created patches (**Fig. 1**) to be 20 and 6 respectively for these models. Having these values greater than 1 reduces the number of time steps substantially and using even greater values for this distance is recommended to be used for sequences that are much longer than 1000 nt. Therefore fairly limited short-term memory of LSTM can be managed. Additionally, it is also possible to change LSTM altogether with transformer-based models, which we will evaluate in the next version. “*

We acknowledge that the selection of hyperparameters can have an impact on the performance of the network and further optimizations could potentially enhance the performance. However, our primary objective was to demonstrate that Self-GenomeNet outperforms self-supervised baselines under similar circumstances, including situations where the same model architecture is employed.

Additionally, in future direction, we explicitly state that the performance could be further improved with different, possibly larger, models (we refer to Discussion Section: “*In*

addition, the performance of Self-GenomeNet can potentially be further enhanced by using different architectural modifications, such as deeper networks, or alternative models for the encoder and context network, which we will evaluate in the next version.”). In future work, we plan to investigate the use of bigger self-supervised models with Self-GenomeNet, which involves predicting varying-length neighbor reverse-complement sequences with a contrastive loss. This exploration may offer insights into enhancing the performance of the model. We also work on another paper, which is submitted as complementary work to Nature Communications Biology, in which we name our method as GenomeNet-Architect. In this paper, we develop an optimization framework for the architecture design of deep learning models on genomics data and we consider including this framework to optimize Self-GenomeNet on a follow-up study.

Q5: There should be a discussion (maybe a short 1-2 lines) on the choice of evaluation metrics. Here, class-balanced accuracy is presented for all experiments. Why not precision/recall/F1 scores? In my opinion, when dealing with medical data in particular, minimizing false positives is more important than achieving greater (balanced) accuracy, which does not place more emphasis on false positives. I am sure that there is a reason for the choice of metric. Making it explicit will help understand the reasoning behind the approach and make it easier to evaluate the proposed model.

We appreciate your comment regarding the choice of evaluation metrics and the need for a discussion on this topic. Our selection of evaluation metrics was based on careful consideration of the specific tasks and datasets involved in our study. We added information regarding the choice of evaluation metrics in the revised manuscript to provide further clarity on our reasoning which reads:

“In all experiments except DeepSEA dataset, we report class-balanced accuracy and not precision/recall/F1 scores, because these metrics put an emphasis on positive samples and also choosing a positive class. However, artificially choosing a positive class is harmful as detection of both classes hold equal importance for phage/non-phage classification and fungi/protozoa classification tasks. For the effector protein prediction task, assigning a positive class is also hard as the number of “effector protein” samples are more in both training, validation, and test set. For our experiments on the DeepSEA dataset, we opted for average PR AUC as a metric, based on the findings of Quang and Xie²⁷, who demonstrated that the sparsity of positive binary targets in this dataset can artificially inflate the ROC AUC and thus PR AUC is a more suitable indicator of performance.”

References

1. Kolesnikov, A., Zhai, X. & Beyer, L. Revisiting Self-Supervised Visual Representation Learning. *2019 IEEE/CVF Conference on Computer Vision and Pattern Recognition (CVPR)* Preprint at <https://doi.org/10.1109/cvpr.2019.00202> (2019).
2. Zhang, R., Isola, P. & Efros, A. A. Colorful Image Colorization. in *Computer Vision – ECCV 2016* 649–666 (Springer International Publishing, 2016).

3. van den Oord, A., Li, Y. & Vinyals, O. Representation Learning with Contrastive Predictive Coding. *arXiv [cs.LG]* (2018).
4. Bachman, P., Hjelm, R. D. & Buchwalter, W. Learning representations by maximizing mutual information across views. *arXiv [cs.LG]* (2019).
5. Chen, T., Kornblith, S., Norouzi, M. & Hinton, G. A simple framework for contrastive learning of visual representations. in *Proceedings of the 37th International Conference on Machine Learning* 1597–1607 (JMLR.org, 2020).
6. Chen, X., Xie, S. & He, K. An empirical study of training self-supervised vision transformers. in *2021 IEEE/CVF International Conference on Computer Vision (ICCV)* 9640–9649 (IEEE, 2021).
7. Zbontar, J., Jing, L., Misra, I., LeCun, Y. & Deny, S. Barlow Twins: Self-Supervised Learning via Redundancy Reduction. in *Proceedings of the 38th International Conference on Machine Learning* (eds. Meila, M. & Zhang, T.) vol. 139 12310–12320 (PMLR, 18--24 Jul 2021).
8. Ren, R., Yin, C. & S-T Yau, S. kmer2vec: A Novel Method for Comparing DNA Sequences by word2vec Embedding. *J. Comput. Biol.* **29**, 1001–1021 (2022).
9. O’Leary, N. A. *et al.* Reference sequence (RefSeq) database at NCBI: current status, taxonomic expansion, and functional annotation. *Nucleic Acids Res.* **44**, D733–45 (2016).
10. Dai, A. M. & Le, Q. V. Semi-supervised sequence learning. *Adv. Neural Inf. Process. Syst.* **28**, (2015).
11. Ciortan, M. & Defrance, M. Contrastive self-supervised clustering of scRNA-seq data. *BMC Bioinformatics* **22**, 280 (2021).
12. Fiannaca, A. *et al.* Deep learning models for bacteria taxonomic classification of metagenomic data. *BMC Bioinformatics* **19**, 198 (2018).
13. de Koning, A. P. J., Gu, W., Castoe, T. A., Batzer, M. A. & Pollock, D. D. Repetitive elements may comprise over two-thirds of the human genome. *PLoS Genet.* **7**, e1002384 (2011).
14. Rolnick, D., Veit, A., Belongie, S. & Shavit, N. Deep Learning is Robust to Massive Label Noise. *arXiv [cs.LG]* (2017).
15. Palazzo, A. F. & Gregory, T. R. The case for junk DNA. *PLoS Genet.* **10**, e1004351 (2014).
16. Géron, A. *Hands-On Machine Learning with Scikit-Learn, Keras, and TensorFlow, 3rd Edition.* (2022).
17. Clark, K., Karsch-Mizrachi, I., Lipman, D. J., Ostell, J. & Sayers, E. W. GenBank. *Nucleic Acids Res.* **44**, D67–72 (2016).
18. Li, J. *et al.* SecReT6: a web-based resource for type VI secretion systems found in bacteria. *Environ. Microbiol.* **17**, 2196–2202 (2015).
19. Zhou, J. & Troyanskaya, O. G. Predicting effects of noncoding variants with deep learning-based sequence model. *Nat. Methods* **12**, 931–934 (2015).
20. Grill, J.-B. *et al.* Bootstrap your own latent: A new approach to self-supervised Learning. *arXiv [cs.LG]* (2020).
21. Liu, H., HaoChen, J. Z., Gaidon, A. & Ma, T. Self-supervised Learning is More Robust to Dataset Imbalance. *arXiv [cs.LG]* (2021).
22. Russakovsky, O. *et al.* ImageNet Large Scale Visual Recognition Challenge. *Int. J. Comput. Vis.* **115**, 211–252 (2015).
23. Parkhi, O. M., Vedaldi, A., Zisserman, A. & Jawahar, C. V. Cats and dogs. in *2012 IEEE Conference on Computer Vision and Pattern Recognition* 3498–3505 (2012).
24. Nilsback, M.-E. & Zisserman, A. Automated Flower Classification over a Large Number of Classes. 722–729 (2008).
25. Krause, J., Deng, J., Stark, M. & Fei-Fei, L. Collecting a large-scale dataset of fine-grained cars. https://pure.mpg.de/rest/items/item_2029263/component/file_2029262/content (2013).
26. Robinson, J., Chuang, C.-Y., Sra, S. & Jegelka, S. Contrastive Learning with Hard Negative

Samples. *arXiv [cs.LG]* (2020).

27. Quang, D. & Xie, X. DanQ: a hybrid convolutional and recurrent deep neural network for quantifying the function of DNA sequences. *Nucleic Acids Res.* **44**, e107 (2016).

REVIEWERS' COMMENTS:

Reviewer #1 (Remarks to the Author):

This is a much improved version of the manuscript. I am satisfied with this version of the manuscript.

There are two minor issues.

On line 84, it was stated that "Self-GenomeNet encodes these two subsequences through a representation network consisting of a convolutional encoder network;" however, on lines 577 and 578, the encoder was mentioned to be a fully connected — not convolutional — layer. Please clarify this point.

In Figure 2, subfigures b and d show the performance using 100% of the labeled datasets; however, subfigure c does not show the performance on 100% of the dataset. Please add these results to subfigure c.

Reviewer #2 (Remarks to the Author):

I would like to thank the authors for their detailed and thorough responses to the comments raised. The response and the revised manuscript have addressed my concerns. I am happy to accept the recommendation of the submitted article.

Responds to Reviewer 1

This is a much improved version of the manuscript. I am satisfied with this version of the manuscript.

We thank the reviewer for recognizing our efforts to enhance the manuscript and for the positive feedback. Additionally, we've made some minor corrections in the method section, specifically fixing typos that occurred when we transferred the equations into the word equation format.

Q1: On line 84, it was stated that "Self-GenomeNet encodes these two subsequences through a representation network consisting of a convolutional encoder network;" however, on lines 577 and 578, the encoder was mentioned to be a fully connected — not convolutional — layer. Please clarify this point.

We understand that the original description may have been unclear, and we appreciate the reviewer's opportunity to clarify this point. Generally, our method utilises a convolutional encoder network. However, in our experiments, we set the kernel size of the convolutional encoder network (layer) to be equivalent to the patch size, which is the number of nucleotides given to the encoder network at once. In this way, the convolutional layer applied to the entire input data is equivalent to a fully connected layer applied to each patch. However, we have now realized that this information, particularly the equivalence of these layers, is likely of little interest to readers and decided not to discuss it in the paper. We have now updated the manuscript in Methods section, "Network Architecture Design" subsection, as follows, which we believe to be clear and convey the necessary information about the network architecture:

"Network Architecture Design

*The particular architectures of the encoder network f_θ and the context network C_ϕ are hyperparameters of the method and can be chosen according to the task at hand. We choose f_θ to be a convolutional layer with 1,024 filters and C_ϕ to be an LSTM layer with 512 units. The kernel size of the convolutional layer is set equal to the patch size, which is the number of nucleotides given to the encoder network (**Fig. 1a**), and the stride value of the convolutional layer is equal to the distance between the starting points of the patches. For experiments trained on 150 nt sequences, the patch size is set to 24, and the stride is set to 6, resulting in 75% overlapping patches. For experiments trained on 1,000 nt sequences, the patch size is set to 40, and the stride to 20, resulting in 50% overlapping patches."*

Q2: In Figure 2, subfigures b and d show the performance using 100% of the labeled datasets; however, subfigure c does not show the performance on 100% of the dataset. Please add these results to subfigure c.

We have included the desired results in Fig. 2c for the completeness of the figure. For the entire labelled dataset (100%), we found that all methods perform very similarly. This is likely because, with 4.4M training samples, the entire dataset is too far from the "data scarce"

setting that our method is envisioned for In addition, we observe that the visual presentation of this sub-figure is improved by using a logarithmic scale on the y-axis when 100% results are present. Consequently, we have made this adjustment to the figure. Please see the updated Fig. 2c below.

Figure 2: Comparison of self-supervised methods in data-scarce settings. *Self-GenomeNet* representations outperform other baseline methods, such as language models²¹ trained by predicting single nucleotides or 3-grams, Contrastive Predictive Coding²², and Contrastive-sc¹⁹, especially when a large fraction of available labels are omitted. We train the models in the datasets without using labels and then successively withhold labeled samples to mimic scenarios where labels are scarce (from 100% of available labeled samples to 0.1%). Each point in the plots is trained separately using the corresponding amount of labeled data. The weights of the context and encoder models are initialized with the training results from the SSL task, but they are trained further (fine-tuned), together with the linear layer, on the new supervised task. The label “Supervised” corresponds to the setting without any pre-training, where the weights are initialized randomly for the supervised task. a) Overview of dataset and tasks used for evaluation. b) The results of the viral dataset for 150 nt sequences, c) the DeepSEA dataset, d) and the viral dataset for 1,000 nt sequences.

Respond to Reviewer 2

I would like to thank the authors for their detailed and thorough responses to the comments raised. The response and the revised manuscript have addressed my concerns. I am happy to accept the recommendation of the submitted article.

We thank the reviewer for acknowledging our efforts to improve our manuscript in the rebuttal time and for the positive recommendation.